# Aggregating Quantitative Relative Judgments:
# From Social Choice to Ranking Prediction

**Yixuan Even Xu**
Carnegie Mellon University
yixuanx@cs.cmu.edu

**Hanrui Zhang**
Chinese University of Hong Kong
hanrui@cse.cuhk.edu.hk

**Yu Cheng**
Brown University
yu_cheng@brown.edu

**Vincent Conitzer**
Carnegie Mellon University
conitzer@cs.cmu.edu

## Abstract

Quantitative Relative Judgment Aggregation (QRJA) is a new research topic in (computational) social choice. In the QRJA model, agents provide judgments on the relative quality of different candidates, and the goal is to aggregate these judgments across all agents. In this work, our main conceptual contribution is to explore the interplay between QRJA in a social choice context and its application to ranking prediction. We observe that in QRJA, judges do not have to be people with subjective opinions; for example, a race can be viewed as a "judgment" on the contestants' relative abilities. This allows us to aggregate results from multiple races to evaluate the contestants' true qualities. At a technical level, we introduce new aggregation rules for QRJA and study their structural and computational properties. We evaluate the proposed methods on data from various real races and show that QRJA-based methods offer effective and interpretable ranking predictions.

## 1 Introduction

In *voting theory*, each voter *ranks* a set of candidates, and a *voting rule* maps the vector of rankings to either a winning candidate or an aggregate ranking of all the candidates. There has been significant interaction between computer scientists interested in voting theory and the *learning-to-rank* community. The learning-to-rank community is interested in problems such as ranking webpages in response to a search query, or ranking recommendations to a user (see, e.g., Liu [2009]). Another problem of interest is to aggregate multiple rankings into a single one, for example combining the ranking results from different algorithms ("voters") into a single meta-ranking. While the interests of the communities may differ, e.g., the learning-to-rank community is less concerned about strategic aspects of voting, a natural intersection point for these two communities is a model where there is a latent "true" ranking of the candidates, of which all the votes are just noisy observations. Consequently, it is natural to try to estimate the true ranking based on the received rankings, and such an estimation procedure corresponds to a voting rule. (See, e.g., Young [1995]; Conitzer and Sandholm [2005]; Meila *et al.* [2007]; Conitzer *et al.* [2009]; Caragiannis *et al.* [2013]; Soufiani *et al.* [2014]; Xia [2016], and Elkind and Slinko [2015] for an overview.)

Voting rules are just one type of mechanism in the broader field of *social choice*, which studies the broader problem of making decisions based on the opinions and preferences of multiple agents. Such opinions are not necessarily represented as rankings. For example, in *judgment aggregation* (see Endriss [2015] for an overview), judges assess whether certain propositions are true or false, and the goal is to aggregate these judgments into logically consistent statements. The observation

38th Conference on Neural Information Processing Systems (NeurIPS 2024).

that other types of input are aggregated in social choice prompts the natural question of whether analogous problems exist in statistics and machine learning (as is the case with ranking aggregation).

In this paper, we focus on a relatively new model in social choice, the *quantitative* judgment aggregation problem [Conitzer *et al.*, 2015, 2016]. In this problem, the goal is to aggregate *relative quantitative judgments*: for example, one agent may value the life of a 20-year-old at 2 times the life of a 50-year-old (say in the context of self-driving cars making decisions) [Noothigattu *et al.*, 2018]; another example could be that an agent judges that "using 1 unit of gasoline is as bad as creating 3 units of landfill trash" (in a societal tradeoff context) [Conitzer *et al.*, 2016]. Quantitative judgment aggregation has been considered in the area of automated moral decision-making, where an AI system may choose a course of action based on data about human judgments in similar scenarios.

An important conceptual difference between this work and previous studies on quantitative judgment aggregation is that we observe that relative "judgments" can be produced by a process other than a subjective agent reporting them, which is the standard assumption in social choice. To illustrate, consider a race in which contestant A finishes at 20:00 and contestant B at 30:00. In this race, the "judgment" is that A is 10:00 faster than B. This key observation allows us to bring the social choice community and the learning-to-rank community closer together, by applying existing social choice formulations of quantitative judgment aggregation to the problem of ranking prediction.

Under this new perspective, the formulation of quantitative judgment aggregation can be applied a set of new scenarios, like ranking contestants using "judgments" from past races, or ranking products based on "judgments" from their sales data. We are interested in aggregating such "judgments" from past data, and using them to predict future rankings. Given the different motivations, some important aspects in a social choice context are less important in our setting. For example, social choice is often concerned with agents strategically misreporting, but this is less relevant in our setting because the "judgments" considered in our setting are not strategic.

**Our Contributions.** We summarize our main contributions below: **(1)** Conceptually, we apply social-choice-motivated solution concepts to the problem of ranking prediction, which creates a bridge between research typically done in the social choice and the learning-to-rank communities. **(2)** We pose and study the problem of quantitative relative judgment aggregation (QRJA) in Section 3, which generalizes models from previous work [Conitzer *et al.*, 2015, 2016]. **(3)** Theoretically, we focus on $\ell_p$ QRJA, an important subclass of QRJA problems. We (almost) settle the computational complexity of $\ell_p$ QRJA in Section 4, proving that $\ell_p$ QRJA is solvable in almost-linear time when $p \geq 1$, and is NP-hard when $p < 1$. **(4)** Empirically, we focus on $\ell_1$ and $\ell_2$ QRJA. We conduct extensive experiments on a wide range of real-world datasets in Section 5 to compare the performance of QRJA with several other commonly used methods, showing the effectiveness of QRJA in practice.

## 2 Motivating Examples

To better motivate our study and help readers understand the problem, we first consider simple mean/median approaches for aggregating quantitative judgments and illustrate their limitations through three examples.

**Example 1.** When each race has some common "difficulty" factor (e.g. how hilly a marathon route is), if a contestant only participates in the "easy" races (or only the "hard" races), simply taking the median or mean of historical performance will return biased estimates, as illustrated in Figure 1.

| Contestant \ Race | Boston | New York | Chicago |
|---|---|---|---|
| Alice | 4:00:00 | 4:10:00 | 3:50:00 |
| Bob | 4:11:00 | 4:18:00 | 4:01:00 |
| Charlie | | | 4:09:00 |

Figure 1: Bob finishes earlier than Charlie in the Chicago race, which suggests that Bob runs marathons faster than Charlie. However, if we simply calculate the mean or median of all available data, Charlie's mean/median finishing time will be faster than Bob's. This is because, Charlie participated only in the Chicago race, where conditions were more favorable.

**Example 2.** Suppose past data shows that Alice has beaten Bob in some race, and Bob has beaten Charlie in another race. If we have never seen Alice and Charlie competing in the same race, we may want to predict that Alice runs faster than Charlie (see Figure 2). However, when comparing Alice

and Charlie, simple measures like median and mean effectively ignore the data on Bob, even though Bob's data can provide useful information for this comparison.

| Contestant \ Race | Boston | New York | Chicago |
|:---:|:---:|:---:|:---:|
| Alice | | 4:10:00 | |
| Bob | 4:11:00 | 4:18:00 | 4:01:00 |
| Charlie | | | 4:09:00 |

Figure 2: The same results as in Figure 1, but with some data missing. If we only look at the data on Alice and Charlie, it is difficult to judge who is the faster runner. If anything, Charlie appears to be slightly faster. However, if we know Bob's results in these races, then transitivity suggests that Alice runs faster than Charlie.

**Example 3.** When the variance of the races' difficulty is much higher than the variance in the contestants' performance, taking the median will essentially focus on the result of a single race (with median difficulty) and may throw away useful information as shown in Figure 3.

| Contestant \ Race | Boston | New York | Chicago |
|:---:|:---:|:---:|:---:|
| Alice | 4:00:00 | 4:10:00 | 3:50:00 |
| Bob | 4:11:00 | 4:18:00 | 4:01:00 |
| Charlie | 4:10:00 | 4:32:00 | 4:09:00 |

Figure 3: In this example, the races' difficulty has high variance, and everyone's median time is in Boston. Based on this, we would predict Charlie to be faster than Bob. However, if we consider the other two races, overall it seems that Bob runs faster than Charlie.

QRJA addresses the above issues by considering *relative* performance instead of absolute performance. More specifically, each race provides a judgment of the form "A runs faster than B by Y minutes" for every pair of contestants $(A, B)$ that participated in this race.

## 3 Problem Formulation

In this section, we formally define the Quantitative Relative Judgment Aggregation (QRJA) problem. We start with the definition of its input.

**Definition 1** (Quantitative Relative Judgment). *For a set of $n$ candidates $N = \{1, \ldots, n\}$, a **quantitative relative judgment** is a tuple $J = (a, b, y)$, denoting a judgment that candidate $a \in N$ is better than candidate $b \in N$ by $y \in \mathbb{R}$ units.*

The input of QRJA is a set of quantitative relative judgments to be aggregated. We model the aggregation result as a vector $\mathbf{x} \in \mathbb{R}^n$, where $x_i$ is the single-dimensional evaluation of candidate $i$. The aggregation result should be consistent with the input judgments as much as possible, i.e., for a quantitative relative judgment $(a, b, y)$, we want $|x_a - x_b - y|$ to be small. We use a loss function $f(|x_a - x_b - y|)$ to measure the inconsistency between the aggregation result and the input judgments. The aggregation result should minimize the weighted total loss. Formally, we define QRJA as follows.

**Definition 2** (Quantitative Relative Judgment Aggregation (QRJA)). *Consider $n$ candidates $N = \{1, \ldots, n\}$ and $m$ quantitative relative judgments $\mathbf{J} = (J_1, \ldots, J_m)$ with weights $\mathbf{w} = (w_1, \ldots, w_m)$ where $J_i = (a_i, b_i, y_i)$. The **quantitative relative judgment aggregation** problem with loss function $f : \mathbb{R}_{\geq 0} \to \mathbb{R}_{\geq 0}$ asks for a vector $\mathbf{x} \in \mathbb{R}^n$ that minimizes $\sum_{i=1}^{m} w_i f(|x_{a_i} - x_{b_i} - y_i|)$.*

Previous work [Conitzer *et al.*, 2015, 2016; Zhang *et al.*, 2019] studied a special case of QRJA where $f(t) = t$. In this work, we broaden the scope and study QRJA with more general loss functions. We first note that when the loss function $f$ is convex, QRJA can be formulated as a convex optimization problem. Consequently, one can use standard convex optimization methods like gradient descent or the ellipsoid method to solve QRJA in polynomial time.

However, general-purpose convex optimization methods are often very slow when the numbers of candidates $n$ and judgments $m$ are large. For this reason, we focus on $\ell_p$ QRJA, an important subclass of QRJA problems with loss function $f(t) = t^p$. Our theoretical analysis (almost) settles the computational complexity of $\ell_p$ QRJA for all $p > 0$. We show that $\ell_p$ QRJA is solvable in

almost-linear time when $p \geq 1$, and is NP-hard when $p < 1$. Our experiments focus on comparing $\ell_1$ and $\ell_2$ QRJA with various baselines in social choice and machine learning. We conduct extensive experiments on a wide range of real-world data sets.

# 4 Theoretical Aspects of $\ell_p$ QRJA

In this section, we study the theoretical aspects of $\ell_p$ QRJA, providing a clean and (almost) tight characterization of the computational complexity of $\ell_p$ QRJA for different values of $p$. Recall that $n$ is the number of candidates and $m$ is the number of judgments. Note that $n \leq 2m$.

In Section 4.1, we prove that for all $p \geq 1$, $\ell_p$ QRJA can be solved in almost-linear time $O(m^{1+o(1)})$. In Section 4.2, we show that when $p < 1$, $\ell_p$ QRJA is NP-hard and there is no FPTAS [1] unless P = NP. Additionally, in Appendix A, we show that if $1 \leq p \leq 2$ and $m \gg n$, we can reduce $m$ to $\widetilde{O}(n)$ while incurring a small error. [2]

## 4.1 $\ell_p$ QRJA in Almost-Linear Time When $p \geq 1$

We first show that when $p \geq 1$, $\ell_p$ QRJA can be solved in $O(m^{1+o(1)})$ time, i.e., in time almost linear in the size of the input. Note that to solve $\ell_p$ QRJA with $p \geq 1$ in polynomial time, one can formulate the problem as an $\ell_p$ regression problem and apply general-purpose techniques for $\ell_p$ regression, e.g., [Bubeck *et al.*, 2018; Adil *et al.*, 2024]. However, these methods would result in a running time that is $\Omega(m + n^\omega)$, where $\omega \geq 2$ is the matrix multiplication exponent. This is significantly slower than almost-linear time. Our approach leverages the additional structure of the QRJA problem, and utilizes the recent advancements in faster algorithms for (directed) maximum flow [Chen *et al.*, 2022].

**Theorem 1.** *Let $p \geq 1$ be an absolute constant. Consider $\ell_p$ QRJA in Definition 2 with loss function $f(t) = t^p$. Assume all input numbers are polynomially bounded in $m$. We can solve $\ell_p$ QRJA in time $O(m^{1+o(1)})$ with $\exp(-\log^c m)$ additive error for any constant $c > 0$.*

**Proof of Theorem 1:** We first prove the theorem for $p > 1$. We will prove the $p = 1$ case in Appendix B.1. Let $S_{\text{input}} = (n, m, (w_i)_{i=1}^m, (y_i)_{i=1}^m)$. We assume $m$ is sufficiently large, and that $c$ is a sufficiently large constant such that $\forall v \in S_{\text{input}}$, either $v = 0$ or $1/m^c < |v| < m^c$.

Consider an $\ell_p$ QRJA instance $(N, \mathbf{J}, \mathbf{w})$ where $\mathbf{J} = (J_1, \ldots, J_m)$ and $J_i = (a_i, b_i, y_i)$, we construct a matrix $\mathbf{A} \in \mathbb{R}^{m \times n}$ and a vector $\mathbf{z} \in \mathbb{R}^m$ as follows:

$$A_{i,j} = \begin{cases} \sqrt[p]{w_i} & \text{if } j = a_i \\ -\sqrt[p]{w_i} & \text{if } j = b_i \\ 0 & \text{otherwise} \end{cases}, \quad z_i = \sqrt[p]{w_i} y_i. \tag{1}$$

Given $\mathbf{A}$ and $\mathbf{z}$, the $\ell_p$ QRJA problem can be formulated as

$$\min_{\mathbf{x} \in \mathbb{R}^n} \sum_{i=1}^m w_i |x_{a_i} - x_{b_i} - y_i|^p = \min_{\mathbf{x} \in \mathbb{R}^n} \|\mathbf{A}\mathbf{x} - \mathbf{z}\|_p^p,$$

We will show how to find $\mathbf{x}$ in time $O(m^{1+o(1)})$ such that

$$\|\mathbf{A}\mathbf{x} - \mathbf{z}\|_p \leq \min_{\mathbf{x}^*} \|\mathbf{A}\mathbf{x}^* - \mathbf{z}\|_p + \exp(-\log^{2c} m).$$

We first write the optimization as

$$\min_{\mathbf{x} \in \mathbb{R}^n} \|\mathbf{A}\mathbf{x} - \mathbf{z}\|_p = \min_{\mathbf{x} \in \mathbb{R}^n, \mathbf{s} \in \mathbb{R}^m, \mathbf{s} = \mathbf{A}\mathbf{x} - \mathbf{z}} \|\mathbf{s}\|_p. \tag{2}$$

The Lagrangian dual of (2) is

$$\min_{\mathbf{x} \in \mathbb{R}^n, \mathbf{s} \in \mathbb{R}^m} \max_{\mathbf{f} \in \mathbb{R}^m} \left( \|\mathbf{s}\|_p + \mathbf{f}^\top (\mathbf{s} - (\mathbf{A}\mathbf{x} - \mathbf{z})) \right).$$

---

[1] Fully Polynomial-Time Approximation Scheme.

[2] The $\widetilde{O}(\cdot)$ notation hides logarithmic factors in its argument.

Note that $\mathbf{s} = \mathbf{A}\mathbf{x} - \mathbf{z}$ is enforced; otherwise the inner maximization problem is unbounded. Let $\|\cdot\|_q$ be the dual norm of $\|\cdot\|_p$, i.e., $\frac{1}{p} + \frac{1}{q} = 1$. (So $q > 1$.) By strong duality,

$$
\begin{aligned}
\max_{\mathbf{f}\in\mathbb{R}^m} \min_{\mathbf{x}\in\mathbb{R}^n,\mathbf{s}\in\mathbb{R}^m} & \left( \|\mathbf{s}\|_p + \mathbf{f}^\top(\mathbf{s} - (\mathbf{A}\mathbf{x} - \mathbf{z})) \right) \\
= \max_{\mathbf{f}\in\mathbb{R}^m} & \left[ \mathbf{f}^\top\mathbf{z} + \min_{\mathbf{s}\in\mathbb{R}^m} \left( \|\mathbf{s}\|_p + \mathbf{f}^\top\mathbf{s} \right) - \max_{\mathbf{x}\in\mathbb{R}^n} \mathbf{f}^\top\mathbf{A}\mathbf{x} \right] \\
= \max_{\mathbf{f}\in\mathbb{R}^m, \mathbf{A}^\top\mathbf{f}=\mathbf{0}, \|\mathbf{f}\|_q\le 1} & \mathbf{f}^\top\mathbf{z}.
\end{aligned}
\tag{3}
$$

The last step follows from the fact that the value of $(\min_{\mathbf{s}\in\mathbb{R}^m} \|\mathbf{s}\|_p + \mathbf{f}^\top\mathbf{s})$ is 0 if $\|\mathbf{f}\|_q \le 1$ and $-\infty$ otherwise, and that $\max_{\mathbf{x}\in\mathbb{R}^n} \mathbf{f}^\top\mathbf{A}\mathbf{x}$ is unbounded if $\mathbf{A}^\top\mathbf{f} \ne \mathbf{0}$.

We will show that the dual program (3) can be solved near-optimally in almost-linear time (Lemma 1), and given a near-optimal dual solution $\mathbf{f} \in \mathbb{R}^m$, a good primal solution $\mathbf{x} \in \mathbb{R}^n$ can be computed in linear time (Lemma 2). Theorem 1 follows directly from Lemmas 1 and 2. ∎

**Lemma 1.** *We can find a feasible solution* $\mathbf{f} \in \mathbb{R}^m$ *of (3) in time* $O(m^{1+o(1)})$ *with additive error* $\exp(-\log^{6c} m)$.

**Proof of Lemma 1:** Consider the following problem, which moves the norm constraint of (3) into the objective:

$$
\max_{\mathbf{f}\in\mathbb{R}^m, \mathbf{A}^\top\mathbf{f}=\mathbf{0}} \mathbf{f}^\top\mathbf{z} - \|\mathbf{f}\|_q^q.
\tag{4}
$$

(4) is closely related to $\ell_p$ norm mincost flow. Recent breakthrough in mincost flow [Chen *et al.*, 2022] showed that a feasible solution $\mathbf{f}^\dagger$ of (4) within error $\exp(-\log^{13c} m)$ can be computed in $O(m^{1+o(1)})$ time.

Suppose $\|\mathbf{f}^\dagger\|_q \ge \exp(-\log^{7c} m)$, which we prove later. Notice that $\mathbf{f}^\dagger$ is a solution within error $\exp(-\log^{13c} m)$ of

$$
\max_{\mathbf{f}\in\mathbb{R}^m, \mathbf{A}^\top\mathbf{f}=\mathbf{0}, \|\mathbf{f}\|_q=\|\mathbf{f}^\dagger\|_q} \mathbf{f}^\top\mathbf{z}.
$$

Choosing $\mathbf{f} = \mathbf{f}^\dagger / \|\mathbf{f}^\dagger\|_q$ satisfies Lemma 1.

To lower bound $\|\mathbf{f}^\dagger\|_q$, let $\mathbf{f}^*$ be the optimal solution of (3). When $\mathbf{f}^{*\top}\mathbf{z} \ge 3$, because the optimal value of (4) is at least $\mathbf{f}^{*\top}\mathbf{z} - 1$ and $\mathbf{f}^\dagger$ is near-optimal for (4), we have $\mathbf{f}^{\dagger\top}\mathbf{z} \ge \mathbf{f}^{*\top}\mathbf{z} - 2$ and thus $\|\mathbf{f}^\dagger\|_q \ge 1/3$. When $\mathbf{f}^{*\top}\mathbf{z} < 3$, we will show $\mathbf{f}^{\dagger\top}\mathbf{z} \ge \exp(-\log^{6c} m)$, so $\|\mathbf{f}^\dagger\|_q \ge \exp(-\log^{7c} m)$.

To show $\mathbf{f}^{\dagger\top}\mathbf{z} \ge \exp(-\log^{6c} m)$, we only need to show that the optimal value of (4) is at least $\exp(-\log^{5c} m)$. We can assume w.l.o.g. that $\mathbf{f}^{*\top}\mathbf{z} > \exp(-\log^{3c} m)$, otherwise there is a primal solution $\mathbf{x}$ almost consistent with all judgments, which is easy to approximate. Note that when scaling down $\mathbf{f}^*$, $\|\mathbf{f}^*\|_q^q$ scales faster than $\mathbf{f}^{*\top}\mathbf{z}$. Let $\mathbf{f}' = k\mathbf{f}^*$ with $k = \exp(-\log^{4c} m)$. We have $\mathbf{f}'^\top\mathbf{z} - \|\mathbf{f}'\|_q^q = k(\mathbf{f}^{*\top}\mathbf{z}) - k^q > \exp(-\log^{5c} m)$, where the last step assumes that $m$ is sufficiently large, in particular $\log^c m > \max\{\frac{2}{q-1}, q+1\}$. ∎

**Lemma 2.** *Given a solution* $\mathbf{f}$ *of (3) that satisfies Lemma 1, we can compute a vector* $\mathbf{x} \in \mathbb{R}^n$ *in time* $O(m)$ *such that*

$$
\|\mathbf{A}\mathbf{x} - \mathbf{z}\|_p \le \min_{\mathbf{x}^*} \|\mathbf{A}\mathbf{x}^* - \mathbf{z}\|_p + \exp(-\log^{2c} m).
$$

**Proof of Lemma 2:** We assume w.l.o.g. that $\|\mathbf{f}\|_q = 1$.

Let $v = \mathbf{f}^\top\mathbf{z}$ and consider

$$
\max_{\mathbf{f}'\in\mathbb{R}^m, \mathbf{A}^\top\mathbf{f}'=\mathbf{0}} \Phi(\mathbf{f}') \quad \text{where} \quad \Phi(\mathbf{f}') = \mathbf{f}'^\top\mathbf{z} - \frac{v}{q} \|\mathbf{f}'\|_q^q.
\tag{5}
$$

Because $\mathbf{f}$ is a solution of (3) within error $\exp(-\log^{6c} m)$, and $\max_{\|\mathbf{f}\|_q} v\,\|\mathbf{f}\|_q - \frac{v}{q}\,\|\mathbf{f}\|_q^q$ is achieved when $\|\mathbf{f}\|_q = 1$, we know that $\mathbf{f}$ is a solution of (5) within error $\exp(-\log^{5c} m)$.

The first-order optimality condition of (5) guarantees that $\nabla\Phi(\mathbf{f})$ is very close to a potential flow. That is, we can find in $O(m)$ time a vector $\mathbf{x} \in \mathbb{R}^n$, such that $\|\mathbf{Ax} - \nabla\Phi(\mathbf{f})\|_\infty \leq \exp(-\log^{3c} m)$. For this $\mathbf{x}$,

$$
\begin{aligned}
\|\mathbf{Ax} - \mathbf{z}\|_p &\leq \|\nabla\Phi(\mathbf{f}) - \mathbf{z}\|_p + \|\mathbf{Ax} - \nabla\Phi(\mathbf{f})\|_p \\
&= v + \|\mathbf{Ax} - \nabla\Phi(\mathbf{f})\|_p \\
&\leq v + m\,\|\mathbf{Ax} - \nabla\Phi(\mathbf{f})\|_\infty \\
&\leq v + \exp(-\log^{2c} m) \\
&\leq \min_{\mathbf{x}^* \in \mathbb{R}^n} \|\mathbf{Ax}^* - \mathbf{z}\|_p + \exp(-\log^{2c} m).
\end{aligned}
$$

The last inequality uses that $v = \mathbf{f}^\top \mathbf{z}$ is a lower bound on the optimal value because $\mathbf{f}$ is a feasible dual solution. ∎

### 4.2 NP-Hardness of $\ell_p$ QRJA When $p < 1$

In this section, we show that $\ell_p$ QRJA is NP-hard when $p < 1$ by reducing from Max-Cut. Note that in this case, the loss function $f(t) = t^p$ is no longer convex.

**Definition 3** (Max-Cut). *For an undirected graph $G = (V, E)$, Max-Cut asks for a partition of $V$ into two sets $S$ and $T$ that the number of edges between $S$ and $T$ is maximized.*

**Reduction from Max-Cut to $\ell_p$ QRJA.** Given a Max-Cut instance on an undirected graph $G = (V, E)$, let $n = |V|, m = |E|, w_2 = \frac{2n}{1-p} + 1$, and $w_1 = nw_2 + 1$.

We will construct an $\ell_p$ QRJA instance with $n + 2$ candidates $V \cup \{v^{(s)}, v^{(t)}\}$ and $O(n + m)$ quantitative relative judgments. Specifically, we add the following judgments:

- $(v^{(t)}, v^{(s)}, 1)$ with weight $w_1$.
- $(v^{(s)}, u, 0)$ with weight $w_2$ for each $u \in V$.
- $(v^{(t)}, u, 0)$ with weight $w_2$ for each $u \in V$.
- $(u, v, 1), (v, u, 1)$ with weight 1 for each $(u, v) \in E$.

In Appendix B.2, we will prove that the Max-Cut instance has a cut of size at least $k$ if and only if the constructed $\ell_p$ QRJA instance has a solution with loss at most $nw_2 + 2(m - k) + k2^p$, which implies the following hardness result.

**Theorem 2.** *For any $p < 1$, there exists a constant $c > 0$ such that it is NP-hard to approximate $\ell_p$ QRJA within a multiplicative factor of $\left(1 + \frac{c}{n^2}\right)$.*

Theorem 2 implies that there is no (multiplicative) FPTAS for $\ell_p$ QJA when $p < 1$ unless P = NP. This is because if a $(1 + \varepsilon)$ solution can be computed in $\mathrm{poly}(m, 1/\varepsilon)$ time, then choosing $\varepsilon = \frac{c}{n^2}$ gives a poly-time algorithm for Max-Cut.

## 5 Experiments

We conduct experiments on real-world datasets to compare the performance of $\ell_1$ and $\ell_2$ QRJA with existing methods. We focus on $\ell_1$ and $\ell_2$ QRJA because the almost-linear time algorithm for general values of $p \geq 1$ relies on very complicated galactic algorithms for $\ell_p$ norm mincost flow [Chen *et al.*, 2022]. Although general-purpose convex optimization methods can also be used to solve $\ell_p$ QRJA, they are not efficient enough for some of the large-scale datasets we use. All experiments are done on a server with 56 CPU cores and 504G RAM. The experiments in Section 5 and Appendices A and C take around 2 weeks in total to run on this server. No GPU is used. All source code is available at
`https://github.com/YixuanEvenXu/quantitative-judgment-aggregation`.

## 5.1 Experiments Setup

**Datasets.** We consider types of contests where events are reasonably frequent (so it makes sense to predict future events based on past ones), and contest results contain numerical scores in addition to rankings. Specifically, we use the four datasets listed below. We include additional experiments on three more datasets in Appendix C, and the copyright information of the datasets in Appendix E.

- **Chess.** This dataset contains the results of the Tata Steel Chess Tournament (`https://tatasteelchess.com/`, also historically known as the Hoogovens Tournament or the Corus Chess Tournament) from 1983 to 2023 [3]. Each contest is typically a round-robin tournament among 10 to 14 contestants. A contestant's numerical score is the contestant's number of wins in the tournament. There are 80 contests and 408 contestants in this dataset.

- **F1.** This dataset contains the results of Formula 1 races (`https://www.formula1.com/`) from 1950 to 2023. In each contest, we take all contestants who complete the whole race. There are around 7 such contestants in each contest. A contestant's numerical score is the negative of his/her finishing time (in seconds). There are 878 contests and 261 contestants in this dataset.

- **Marathon.** This dataset contains the results of the Boston and New York Marathons from 2000 to 2023. We use the data from `https://www.marathonguide.com/`, which publishes results of all major marathon events. Each contest usually involves more than 20000 contestants. We take the 100 top-ranked contestants in each contest as our dataset. A contestant's numerical score is the negative of that contestant's finishing time (in seconds). There are 44 contests and 2984 contestants.

- **Codeforces.** This dataset contains the results of Codeforces (`https://codeforces.com`), a website hosting frequent online programming contests, from 2010 to 2023 (Codeforces Round 875). We consider only Division 1 contests, where only more skilled contestants can participate. Each contest involves around 700 contestants. We take the 100 top-ranked contestants in each contest as our dataset. A contestant's numerical score is that contestant's points in that contest. There are 327 contests and 5338 contestants in total in this dataset.

**Evaluation Metrics.** For all the datasets we use, contests are naturally ordered chronologically. We use the results of the first $i-1$ contests to predict the results of the $i$-th contest. We apply the following two metrics to evaluate the prediction performance of different algorithms.

- **Ordinal Accuracy.** This metric measures the percentage of correct relative ordinal predictions. For each contest, we predict the ordinal results of all pairs of contestants that (i) have both appeared before and (ii) have different numerical scores in the current contest. We compute the percentage of correct predictions.

- **Quantitative Loss.** This metric measures the average absolute error [4] of relative quantitative predictions. For each contest, we predict the difference in numerical scores of all pairs of contestants that have both appeared before. We then compute the quantitative loss as the average absolute error of the predictions. We normalize this number by the quantitative loss of the trivial prediction that always predicts 0 for all pairs.

**Implementation.** We have implemented both $\ell_1$ and $\ell_2$ QRJA in Python. We use Gurobi Gurobi Optimization, LLC [2023] and NetworkX Hagberg *et al.* [2008] to implement $\ell_1$ QRJA and the least-square regression implementation in SciPy [Jones *et al.*, 2014] to implement $\ell_2$ QRJA. To transform the contest standings into a QRJA instance, we construct a quantitative relative judgment $J = (a, b, y)$ for each contest and each pair of contestants $(a, b)$ with $y$ being the score difference between $a$ and $b$ in that contest. We set all weights to 1 to ensure fair comparison with benchmarks.

**Benchmarks.** We evaluate $\ell_1$ and $\ell_2$ QRJA against several benchmark algorithms. Specifically, we consider the natural one-dimensional aggregation methods Mean and Median, social choice methods Borda and Kemeny-Young, and a common method for prediction, matrix factorization. We describe how we apply these methods to our setting below.

---

[3] We choose the time frame of our datasets to be longer than the active period of most contestants to emphasize that contestants come and go, but their past performance could help the prediction.

[4] We also include the experiment results using average squared error as the quantitative metric in Appendix C.1. The relative performance of the tested algorithms on these two metrics are similar.

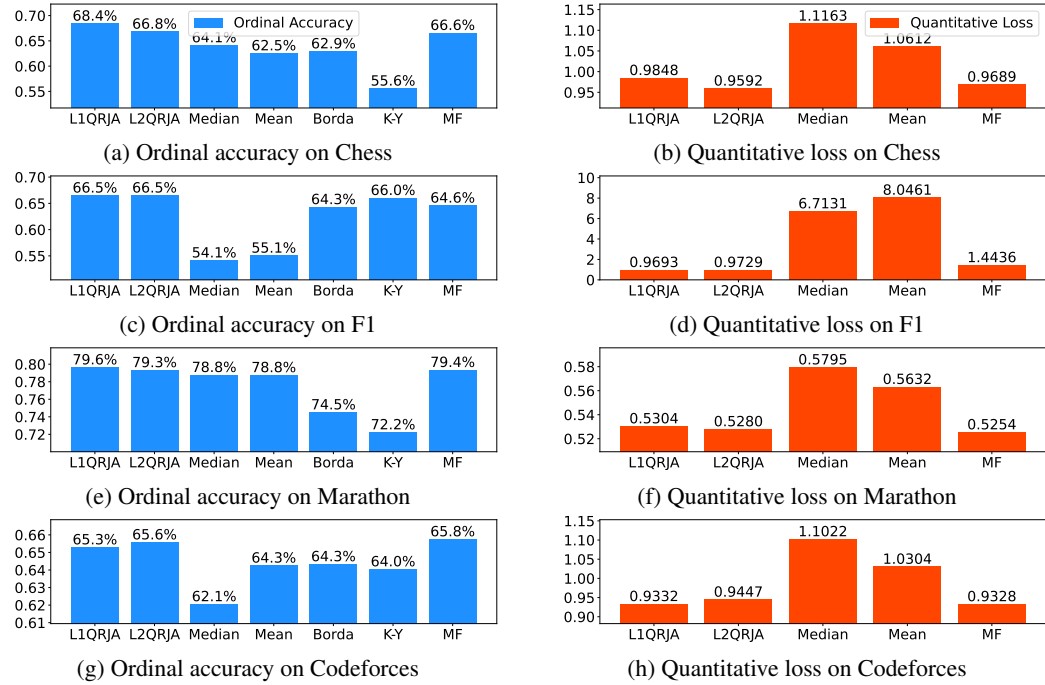

Figure 4: Ordinal accuracy and quantitative loss of the algorithms on all four datasets. Error bars are not shown here as the algorithms are deterministic. The results show that both versions of QRJA perform consistently well across the tested datasets.

- **Mean** and **Median.** For every contestant in the training set, we take the mean or median of that contestant's scores in training contests. We then make predictions based on differences between these mean or median scores. In one-dimensional environments like ours, means and medians are considered to be among the best imputation methods for various tasks (see, e.g., Engels and Diehr, 2003, Shrive *et al.*, 2006).

- **The Borda rule.** The Borda rule is a voting rule that takes rankings as input and produces a ranking as output. We use a normalized version of the Borda rule. The $i$-th ranked contestant in contest $j$ receives $1 - \frac{2(i-1)}{n_j-1}$ points, where $n_j$ is the number of contestants in the contest. The aggregated ranking result is obtained by sorting the contestants by their total number of points.

- **The Kemeny-Young rule.** [Kemeny, 1959; Young and Levenglick, 1978; Young, 1988]. The Kemeny-Young rule is a voting rule that takes multiple (partial) rankings of the contestants as input and produces a ranking as output. Specifically, it outputs a ranking that minimizes the number of *disagreements* on pairs of contestants with the input rankings. Finding the optimal Kemeny-Young ranking is known to be NP-hard Bartholdi *et al.* [1989]. In our experiments, we use Gurobi to solve the mixed-integer program formulation of the Kemeny-Young rule given in Conitzer *et al.* [2006]. As this method is still computationally expensive and can only scale to hundreds of contestants, for each contest we predict, we only keep the contestants within that specific contest and discard all other contestants to run Kemeny-Young.

- **Matrix Factorization (MF).** Matrix factorization takes as input a matrix with missing entries and outputs a prediction of the whole matrix. Every row is a contestant and every column is a race. The score of a contestant in a race is the entry in the corresponding row and column. We implement several variants of MF and report results for one variant (Koren *et al.* [2009]), as other variants have comparable or worse performance. For implementation details and other variants, see Appendix C.4.

Many other, related approaches deserve mention in this context. But we do not include them in the benchmarks because they do not exactly fit our setting or motivation. For example, the seminal Elo rating system Elo [1978] as well as many other methods Maher [1982]; Karlis and Ntzoufras [2008]; Guo *et al.* [2012]; Hunter and others [2004] can all predict the results of pairwise matches in, e.g.,

chess and football. However, they are not originally designed for predicting the results of contests with more than two contestants.

## 5.2 Experiment Results

The complete experimental results of all algorithms on the four datasets are shown in Fig. 4. Note that Borda and Kemeny-Young do not make quantitative predictions, so they are not included in Figs. 4b, 4d, 4f and 4h.

**The performance of QRJA.** As shown in Fig. 4, both versions of QRJA perform consistently well across the tested datasets. They are always among the best algorithms in terms of both ordinal accuracy and quantitative loss.

**The performance of Mean and Median.** In terms of ordinal accuracy, Mean and Median do well on Marathon, but are not among the best algorithms on other datasets, especially on F1 (for both) and Codeforces (for Median). Moreover, for quantitative loss, they are never among the best algorithms.

**The performance of Borda and Kemeny-Young.** Borda and Kemeny-Young do not make quantitative predictions, so we only compare them with other algorithms in terms of ordinal accuracy. As shown in Fig. 4, Borda and Kemeny-Young perform very well on F1, but are not among the best algorithms on other datasets. By only using rankings as input, Borda and Kemeny-Young are more robust on datasets where contestants' performance varies a lot. However, they fail to utilize the quantitative information on other datasets.

**The performance of Matrix Factorization (MF).** MF works well across the tested datasets in terms of both metrics. In all of our four datasets, it has performance comparable to QRJA. The advantage of QRJA over MF is the interpretability of its model. The variables in QRJA have clear meanings - they can be interpreted as the strength of each contestant - in contrast to the latent factors and features in MF, which are harder to interpret. Additionally, we observe in Appendix C.2 that $\ell_1$ QRJA is more robust to large variance in contestants' performance than MF.

**Summary of experimental results.** In summary, both MF and QRJA are never significantly worse than the best-performing algorithm on any of the tested datasets, unlike the other benchmark methods. QRJA additionally offers an interpretable model. This shows that QRJA is an effective method for making predictions on contest results.

## 6 Related Work

**Random utility models.** Random utility models (Fahandar *et al.* [2017]; Zhao *et al.* [2018]) explicitly reason about the contestants being numerically different from each other, e.g., one contestant is generally 1.1 times as fast as another. However, they are still designed for settings in which the only input data we have is ranking data, rather than numerical data such as finishing times. Moreover, random utility models generally do not model common factors, such as a given race being tough and therefore resulting in higher finishing times for *everyone*.

**Matrix completion.** Richer models considered in recommendation systems appear too general for the scenarios we have in mind. Matrix completion Rennie and Srebro [2005]; Candès and Recht [2009] is a popular approach in collaborative filtering, where the goal is to recover missing entries given a partially-observed low-rank matrix. While using higher ranks may lead to better predictions, we want to model contestants in a single-dimensional way, which is necessary for interpretability purposes (the single parameter being interpreted as the "quality" of the contestant).

**Preference learning.** In preference learning, we train on a subset of items that have preferences toward labels and predict the preferences for all items (see, e.g., Pahikkala *et al.* [2009]). One high-level difference is that preference learning tends to use existing methodologies in machine learning to learn rankings. In contrast, our methods (as well as those in previous work Conitzer *et al.* [2015, 2016]) are social-choice-theoretically well motivated. In addition, our methods are designed for quantitative predictions, while the main objective of preference learning is to learn ordinal predictions.

**Elo and TrueSkill.** Empirical methods, such as the Elo rating system Elo [1978] and Microsoft's TrueSkill Herbrich *et al.* [2006], have been developed to maintain rankings of players in various forms of games. Unlike QRJA, these methods focus more on the online aspects of the problem, i.e.,

how to properly update scores after each game. While under specific statistical assumptions, these methods can in principle predict the outcome of a future game, they are not designed for making ordinal or quantitative predictions in their nature.

# 7  Conclusion

In this paper, we conduct a thorough investigation of QRJA (Quantitative Relative Judgment Aggregation). We pose and study QRJA and focus on an important subclass of problems, $\ell_p$ QRJA. Our theoretical analysis shows that $\ell_p$ QRJA can be solved in almost-linear time when $p \geq 1$, and is NP-hard when $p < 1$. Empirically, we conduct experiments on real-world datasets to show that QRJA-based methods are effective for predicting contest results. As mentioned before, the almost-linear time algorithm for general values of $p \neq 1, 2$ relies on very complicated galactic algorithms. An interesting avenue for future work would be to develop fast (e.g., nearly-linear time) algorithms for $\ell_p$ QRJA with $p \neq 1, 2$ that are more practical, and evaluate their empirical performance.

**Broader Impacts.** We expect our work to have a mostly positive social impact by providing an effective and interpretable method for aggregating quantitative relative judgments that can be used in applications such as predicting contest results. While for specific applications, certain desiderata may be not met by QRJA, we allow users (e.g., contest organizers) to set different weights for different judgments, which can be used to reflect the importance of different contests.

## Acknowledgments and Disclosure of Funding

Zhang and Conitzer are supported by NSF IIS-1814056, the Center for Emerging Risk Research, and the Cooperative AI Foundation. Cheng is supported in part by NSF Award CCF-2307106.

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

# A Subsampling Judgments

## A.1 Subsampling Judgments When $p \in [1, 2]$

In this section, we show that for $p \in [1, 2]$, we can reduce the number of judgments while incurring a small approximation error by subsampling the input judgments.

---

**Algorithm 1** Subsampling Judgments

---

**Input**: $\ell_p$ QRJA instance $(N, \mathbf{J}, \mathbf{w})$, subsample count $M \in \mathbb{N}$, and subsampling weights $\mathbf{s} \in \mathbb{R}^m$.
**Output**: $\ell_p$ QRJA instance $(N, \mathbf{J}', \mathbf{w}')$.
 1: Let $q_i \leftarrow \frac{s_i}{\sum_{j=1}^{m} s_j}$ for each $i \in \{1, 2, \dots, m\}$.
 2: **for** $i \in \{1, 2, \dots, M\}$ **do**
 3:     Sample $x \in \{1, 2, \dots, m\}$ with probability $q_x$.
 4:     Let $J_i' \leftarrow J_x$ and $w_i' \leftarrow \frac{w_x}{M \cdot q_x}$.
 5: **end for**
 6: **return** $(N, \mathbf{J}', \mathbf{w}')$.

---

Algorithm 1 takes as input an $\ell_p$ QRJA instance, a parameter $M$, and a vector $\mathbf{s} \in \mathbb{R}^m$. It then samples $M$ judgments from the input instance (with replacements) with probability proportional to $\mathbf{s}$, and outputs a new $\ell_p$ QRJA instance with the sampled judgments. The weight of any judgment in the output instance is divided by its expected number of occurrences in the output instance, so that the expected total weight of any judgment is preserved after subsampling.

**Theorem 3.** *Fix absolute constants $p \in [1, 2]$ and $\varepsilon > 0$. Given any $\ell_p$ QRJA instance $(N, \mathbf{J}, \mathbf{w})$, we can compute subsampling weights $\mathbf{s} \in \mathbb{R}^m$ in time $O(m + n^{\omega + o(1)})$, where $\omega$ is the matrix multiplication exponent. For these weights $\mathbf{s}$ and $M = \widetilde{O}(n)$, Algorithm 1 with high probability outputs an $\ell_p$ QRJA instance $(N, \mathbf{J}', \mathbf{w}')$ whose optimal solution is an $(1 + \varepsilon)$-approximate solution of the original instance.*

To obtain the theoretical guarantee of Algorithm 1, we use the Lewis weights mentioned in (Cohen and Peng [2015]) as vector $\mathbf{s}$. Empirically, we also find that simply setting $\mathbf{s}$ as an all-ones vector works well in many real-world datasets (see Appendix A.2).

**Proof of Theorem 3:** For an $\ell_p$ QRJA instance $(N, \mathbf{J}, \mathbf{w})$, define matrix $\mathbf{A} \in \mathbb{R}^{m \times (n+1)}$

$$
A_{i,j} = \begin{cases}
\sqrt[p]{w_i} & \text{if } j = a_i \\
-\sqrt[p]{w_i} & \text{if } j = b_i \\
-\sqrt[p]{w_i} y_i & \text{if } j = n + 1 \\
0 & \text{otherwise.}
\end{cases}
$$

The **Lewis weights** for this $\ell_p$ QRJA instance is defined as the unique vector $\mathbf{s} \in \mathbb{R}^m$ such that for each $i \in \{1, 2, \dots, m\}$,

$$
\mathbf{a}_i \left( \mathbf{A}^\top \mathbf{S}^{1 - \frac{2}{p}} \mathbf{A} \right)^{-1} \mathbf{a}_i^\top = s_i^{2/p},
$$

where $\mathbf{S} = \text{diag}(\mathbf{s})$ and $\mathbf{a}_i$ is the $i$-th row of $\mathbf{A}$.

The existence and uniqueness of such weights are first shown in Lewis [1978]. In Cohen and Peng [2015], the authors show that for $p \in [1, 2]$, the Lewis weights can be computed in $O(\text{nnz}(\mathbf{A}) + n^{\omega + o(1)}) = O(m + n^{\omega + o(1)})$ time.

For $\mathbf{x} \in \mathbb{R}^n$, we have

$$
\left\| \mathbf{A} \begin{bmatrix} \mathbf{x} \\ 1 \end{bmatrix} \right\|_p^p = \sum_{i=1}^{m} w_i |x_{a_i} - x_{b_i} - y_i|^p.
$$

Thus the $\ell_p$ QRJA loss is always equal to $\|\mathbf{A}\mathbf{x}\|_p^p$ for some $\mathbf{x} \in \mathbb{R}^{n+1}$. The theorem then follows from the $\ell_p$ Matrix Concentration Bounds in Cohen and Peng [2015]. ∎

## A.2 Subsampling Experiments

We also conduct experiments to test the performance of our subsampling algorithm (Algorithm 1), which speeds up the (approximate) computation of QRJA on large datasets. In the experiments, we specify the subsample rate $\alpha$, let $M = \lfloor \alpha m \rfloor$ and $\mathbf{s}$ be an all-ones vector in Algorithm 1.

**Experiment setup.** We run $\ell_1$ and $\ell_2$ QRJA with instances subsampled by Algorithm 1 on the datasets. For each $\alpha = \{0.1, 0.2, \ldots, 1.0\}$, we run $\ell_1$ and $\ell_2$ QRJA 10 times and report their average performance on both metrics with error bars. Due to the space constraints, we only show the results on Chess in Fig. 5 in this section. The results on other datasets are deferred to Appendix C.3.

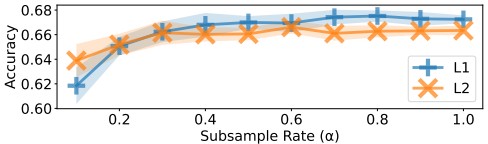 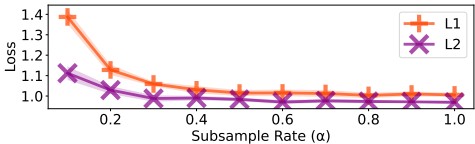

(a) $\ell_1$ and $\ell_2$ QRJA's ordinal accuracy on Chess    (b) $\ell_1$ and $\ell_2$ QRJA's quantitative loss on Chess

Figure 5: The performance of $\ell_1$ and $\ell_2$ QRJA on Chess after subsampling judgments using Algorithm 1 with equal weights for all judgments. The subsample rate $\alpha$ means $M = \lfloor \alpha m \rfloor$ in Algorithm 1. Error bars indicate the standard deviation. The results show that Algorithm 1 can reduce the number of judgments to a factor of $0.4$ with a minor performance loss on Chess.

**Experiment results.** As is shown in Fig. 5, with equal weights for all judgments, Algorithm 1 can reduce the number of judgments without significantly hurting the performance of $\ell_1$ and $\ell_2$ QRJA as long as the sampling rate $\alpha$ is not too small ($\geq 0.4$ for Chess). This shows that Algorithm 1 is a practical algorithm for subsampling judgments in QRJA. We also note that as the experiments show, $\ell_2$ QRJA is more robust to subsampling than $\ell_1$ QRJA.

# B  Missing Proofs in Section 4

## B.1  Proof of Theorem 1

**Theorem 1.** *Let $p \geq 1$ be an absolute constant. Consider $\ell_p$ QRJA in Definition 2 with loss function $f(t) = t^p$. Assume all input numbers are polynomially bounded in $m$. We can solve $\ell_p$ QRJA in time $O(m^{1+o(1)})$ with $\exp(-\log^c m)$ additive error for any constant $c > 0$.*

**Proof of Theorem 1 (when $p = 1$):** We proved Theorem 1 for $p > 1$ in Section 4.1. It remains to consider $p = 1$.

When $p = 1$, the overall loss function of QRJA is a sum of absolute values of some linear terms. We can therefore formulate $\ell_1$ QRJA as the following linear program (LP), as observed in [Zhang *et al.*, 2019]:

$$
\begin{aligned}
\text{minimize} \quad & \sum_{i=1}^{m} w_i \left( z_i^+ + z_i^- \right) \\
\text{subject to} \quad & z_i^+ \geq x_{a_i} - x_{b_i} - y_i && \forall i \in [m] \\
& z_i^- \geq y_i + x_{b_i} - x_{a_i} && \forall i \in [m] \\
& z_i^+ \geq 0, z_i^- \geq 0 && \forall i \in [m] \\
& x_i \in \mathbb{R} && \forall i \in [n]
\end{aligned}
$$

For this LP, Zhang *et al.* [2019] gave a faster algorithm than using general-purpose LP solvers.

**Lemma 3** (Zhang *et al.* 2019). *There is a reduction from $\ell_1$ QRJA to Minimum Cost Flow with $O(n)$ vertices and $O(m)$ edges in $O(T_{\text{SSSP}}(n, m, W))$ time, where $T_{\text{SSSP}}(n, m, W)$ is the time required to solve Single-Source Shortest Path with negative weights on a graph with n vertices, m edges, and maximum absolute distance W.*

Using this reduction (Lemma 3) together with the SSSP algorithm in Bernstein *et al.* [2022] and the minimum cost flow algorithm in Chen *et al.* [2022], we have an algorithm for $\ell_1$ QRJA that runs in time $O(m^{1+o(1)})$. ∎

## B.2 Proof of Theorem 2

**Theorem 2.** *For any $p < 1$, there exists a constant $c > 0$ such that it is NP-hard to approximate $\ell_p$ QRJA within a multiplicative factor of $\left(1 + \frac{c}{n^2}\right)$.*

Recall the reduction from Max-Cut to $\ell_p$ QRJA: Given an instance of Max-Cut with an undirected graph $G = (V, E)$, let $n = |V|, m = |E|$ and let $w_2 = \frac{2n}{1-p} + 1, w_1 = nw_2 + 1$. We construct an instance of $\ell_p$ QRJA with $n + 2$ candidates $V \cup \{v^{(s)}, v^{(t)}\}$ and $O(n + m)$ quantitative relative judgments. Specifically, we construct the followings judgments:

- $(v^{(t)}, v^{(s)}, 1)$ with weight $w_1$.
- $(v^{(s)}, u, 0)$ with weight $w_2$ for each $u \in V$.
- $(v^{(t)}, u, 0)$ with weight $w_2$ for each $u \in V$.
- $(u, v, 1), (v, u, 1)$ with weight 1 for each $(u, v) \in E$.

To show validity of the reduction above, we will first establish integrality of any optimal solution.

**Lemma 4.** *Any optimal solution of the $\ell_p$ QRJA instance described in the above reduction is integral. Moreover, all variables must be either 0 or 1 up to a global constant shift.*

We need an inequality for the proof of Lemma 4.

**Lemma 5.** *For any $d \in (0, \frac{1}{2}], p \in (0, 1)$,*

$$1 - (1 - d)^p \le pd^p.$$

**Proof of Lemma 5:** Fix $p \in (0, 1)$. Let $f(d) = pd^p - 1 + (1 - d)^p$. We have

$$f'(d) = p(pd^{p-1} - (1 - d)^{p-1}).$$

Note that $f'$ is decreasing for $d \in (0, 1)$. In other words, $f$ is single peaked on $(0, \frac{1}{2}]$ and continuous at 0. Now we only have to check that $f(0) \ge 0$, which is trivial, and $f\left(\frac{1}{2}\right) \ge 0$. For the latter, let

$$g(p) = (p + 1)0.5^p - 1.$$

$g(p) \ge 0$ for $p \in [0, 1]$ since $g(p)$ is concave on $[0, 1]$ and $g(0) = g(1) = 0$. The lemma then follows. ∎

We then proceed to prove Lemma 4.

**Proof of Lemma 4:** Let $x_a$ be the potential of candidate $a$ in $\ell_p$ QRJA. W.l.o.g. assume that in any solution, $x_{v^{(s)}} = 0$. We first show that if $x_{v^{(t)}} \ne 1$, then moving it to 1 strictly improves the solution. Suppose $|x_{v^{(t)}} - 1| = d$. By moving $x_{v^{(t)}}$ to 1, we decrease the loss on the judgment $(v^{(t)}, v^{(s)}, 1)$ by $w_1 d^p$. For other judgments $(v^{(t)}, u)$ incident on $v^{(t)}$, the loss increase by no more than $w_2 d^p$, since

$$|(x_{v^{(t)}} \pm d) - x_u|^p \le |x_{v^{(t)}} - x_u|^p + d^p.$$

Overall, the cost decreases by at least

$$w_1 d^p - nw_2 d^p = d^p > 0.$$

Now we show moving any fractional $x_u$ to the closest value in $\{0, 1\}$ strictly improves the solution. There are two cases:

- $x_u \in (0, 1)$. W.l.o.g. $x_u \in (1, \frac{1}{2}]$ and we try to move it to 0 by a displacement of $d = x_u$. The total loss on $(v^{(s)}, u, 0)$ and $(v^{(t)}, u, 0)$ decreases by $w_2(d^p + (1 - d)^p - 1)$, while the total cost on judgments of form $(u, v, 1)$ and $(v, u, 1)$ can increase by no more than $n(d^p + (2 + d)^p - 2^p)$. With Lemma 5, we see that

$$\begin{aligned} w_2(d^p + (1 - d)^p - 1) &\ge w_2(d^p - pd^p) \\ &> 2nd^p \\ &\ge n(d^p + (2 + d)^p - 2^p). \end{aligned}$$

So, there is a positive improvement from rounding $x_u$.

- $x_u \notin [0,1]$. W.l.o.g. $x_u < 0$ and we try to move it to $0$ by a displacement of $d = -x_u$. The total loss on $(v^{(s)}, u, 0)$ and $(v^{(t)}, u, 0)$ decreases by $w_2(d^p + (1+d)^p - 1)$, while the total cost on edges of form $(u, v, 1)$ and $(v, u, 1)$ can increase by no more than $n(d^p + (2+d)^p - 2^p)$. And

$$
\begin{aligned}
w_2(d^p + (1+d)^p - 1) &\geq w_2 d^p \\
&> 2n d^p \\
&\geq n(d^p + (2+d)^p - 2^p).
\end{aligned}
$$

We conclude that in any optimal solution, $x_{v^{(s)}} = 0$, $x_{v^{(t)}} = 1$, and for any $u \in V$, $x_u \in \{0, 1\}$. ∎

Next, we present a lemma that shows the connection between solutions in the Max-Cut instance and those in the constructed $\ell_p$ QRJA instance.

**Lemma 6.** *A Max-Cut instance has a solution of size at least $k$ iff its corresponding $\ell_p$ QRJA instance has a solution of loss at most $n w_2 + 2(m - k) + k2^p$. Moreover, with such a solution to the $\ell_p$ QRJA instance, one can construct a Max-Cut solution of the claimed size.*

**Proof of Lemma 6:** Given a Max-Cut solution $(S, T)$ of size at least $k$, setting the potentials of the vertices in $S$ and $T$ to be $0$ and $1$ respectively gives an $\ell_p$ QRJA solution with loss at most $n w_2 + 2(m - k) + k2^p$.

Given a $\ell_p$ QRJA solution of loss at most $n w_1 + 2(m - k) + k2^p$, we first round the solution to the form stated in Lemma 4. This improves the solution. The two vertex sets $U = \{u \in V \mid x(u) = 0\}$ and $V = \{v \in V \mid x(v) = 1\}$ then form a Max-Cut solution of size at least $k$. ∎

We are now ready to prove Theorem 2.

**Proof of Theorem 2:** According to Lemma 6, any approximation with an additive error less than $2 - 2^p$ of the constructed $\ell_p$ QRJA instance can be rounded to produce an optimal solution to Max-Cut. Since Max-Cut is NP-Hard and the constructed $\ell_p$ QRJA instance's optimal solution has loss $\Theta(n^2 + m)$, the theorem follows. ∎

## C    Additional Experiments

### C.1    L2 Variant of Quantitative Loss

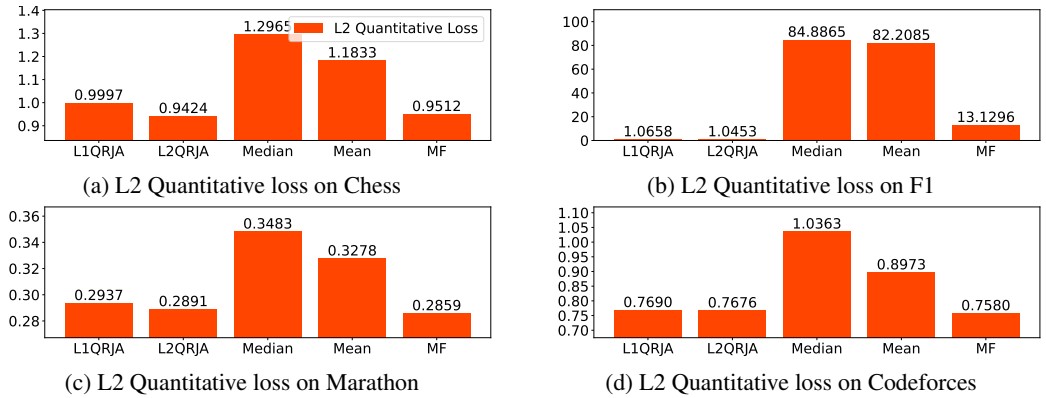

(a) L2 Quantitative loss on Chess

(b) L2 Quantitative loss on F1

(c) L2 Quantitative loss on Marathon

(d) L2 Quantitative loss on Codeforces

Figure 6: L2 quantitative loss of the algorithms on all four datasets used in Section 5. Error bars are not shown here as the algorithms are deterministic. Similar to Fig. 4, the results show that both versions of QRJA perform consistently well across the tested datasets.

We include in this subsection experiment results using average squared error as the quantitative metric. We call this metric **L2 quantitative loss**. Specifically, for each contest, we predict the difference in numerical scores of all pairs of contestants that have both appeared before. We then compute the L2 quantitative loss as the average squared error of the predictions, and normalize it by the L2 quantitative loss of the trivial prediction that always predicts 0 for all pairs.

The results are shown in Fig. 6. We observe that both versions of QRJA still perform consistently well compared to other algorithms across the tested datasets. This is consistent with the results using the (L1) quantitative loss in Section 5.

Additionally, $\ell_2$ QRJA performs slightly better than $\ell_1$ QRJA on this metric. This is expected because this metric is more aligned with the $\ell_2$ QRJA's loss function.

## C.2 Performance Experiments on More Datasets

We include in this subsection the performance experiments on three more datasets. The new datasets are listed below.

- **Cross-Tables.** This dataset contains the results of cross-tables (a crossword-style word game) tournaments (`https://www.cross-tables.com/`) from 2000 to 2023. Each contest is a round-robin tournament involving around 8 contestants. A contestant's numerical score is his/her number of wins in the tournament. There are 1215 contests and 1912 contestants in this dataset.

- **F1-Full.** This dataset is an alternative version of F1. In F1-Full, we choose to additionally include contestants who do not complete the whole race. Now the contestants are ranked first by the number of laps they finish, and then their finishing time. A contestant's numerical score is the negative of the contestant's finishing time (in seconds). If the contestant does not finish all laps, we add a large penalty (1000 seconds) for each lap the contestant fails to finish. There are 878 contests and 606 contestants in this dataset.

- **Codeforces-Core.** This dataset is a modified version of Codeforces. We only keep contestants who have participated in at least half of the contests in this dataset. We test on this modified dataset because all other datasets we use in the experiments are sparse datasets (i.e., contestants participate in a small fraction of the contests on average), so we want to see what happens on dense ones. There are 327 contests and 17 contestants in total.

We evaluate $\ell_1$ and $\ell_2$ QRJA using the same metrics against the same set of benchmarks as in Section 5 on these three datasets. The results are shown in Fig. 7. We highlight a few extra observations below.

**Extra observations on Cross-Tables.** In terms of ordinal accuracy, Median performs the best among the tested algorithms on Cross-Tables. However, in terms of quantitative loss, Median is the worst algorithm among the tested ones. Moreover, it mostly performs suboptimally on other datasets as shown in Figs. 4 and 7. This shows that although Median is occasionally good in performance, it fails in other cases.

**Extra observations on F1-Full.** On F1-Full, both MF and $\ell_2$ QRJA and perform considerably worse than $\ell_1$ QRJA. This is not seen in other datasets. We believe this is because our score calculation results in a large variance in contestants' scores on F1-Full, which makes it harder for these methods to make good predictions. This also shows that $\ell_1$ QRJA is more robust to datasets with large variances in contestants' performance than these methods. We also notice that Borda and Kemeny-Young perform well on F1-Full, which is consistent with their good performance on F1.

**Extra observations on Codeforces-Core.** In terms of ordinal accuracy, all tested algorithms except Borda perform well. In terms of quantitative loss, MF and Median are worse than the other ones. This shows that on a dense dataset like Codeforces-Core, most algorithms can make good predictions. Moreover, MF does not have a clear advantage over other algorithms in our problem even if the dataset is dense.

## C.3 Subsampling Experiments on More Datasets

We also conduct the subsampling experiments in Appendix A.2 on all other 5 datasets. The results are shown in Fig. 8.

**Experiment results.** The message here is the same as that in Appendix A.2. In particular, Algorithm 1 can reduce the number of judgments with only a minor loss in performance as long as the subsample rate $\alpha$ is not too small. Note that in some of the figures, like Fig. 8c, the errors seem to be large visually. This is because of the small scale of the y-axis (only 0.6% for Fig. 8c). The actual errors are

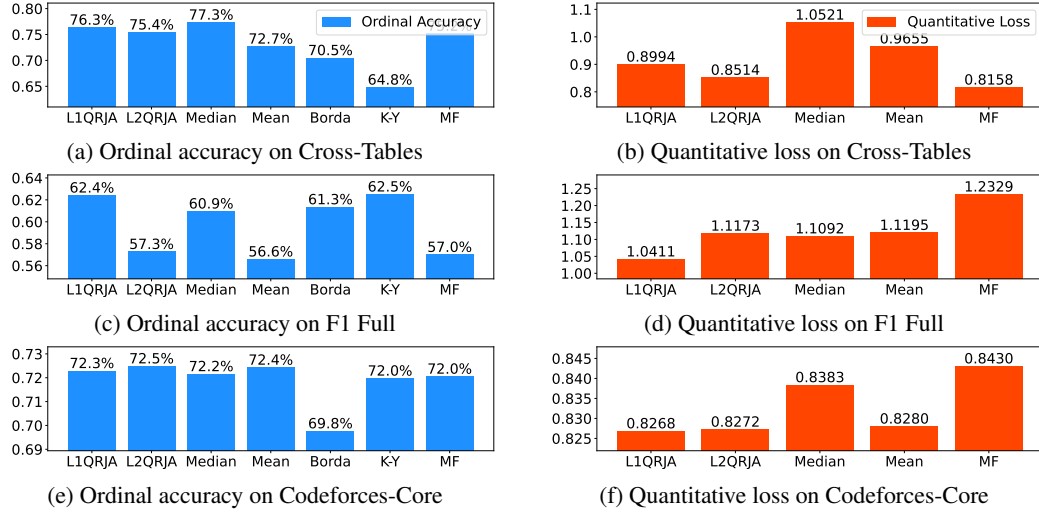

(a) Ordinal accuracy on Cross-Tables

(b) Quantitative loss on Cross-Tables

(c) Ordinal accuracy on F1 Full

(d) Quantitative loss on F1 Full

(e) Ordinal accuracy on Codeforces-Core

(f) Quantitative loss on Codeforces-Core

Figure 7: The performance of the algorithms on Cross-Tables, F1-Full, and Codeforces-Core. Error bars are not shown as the algorithms are deterministic. The results show that $\ell_1$ QRJA still performs consistently well across the tested datasets. However, $\ell_2$ QRJA performs considerably worse than $\ell_1$ QRJA on F1-Full. This is not seen in other datasets.

small. Moreover, we observe that the performance of $\ell_2$ QRJA is slightly more robust to subsampling than that of $\ell_1$ QRJA. This is consistent with the results in Appendix A.2.

## C.4 Experiments about Matrix Factorization

Recall that in Section 5, we only show results of one version of Matrix Factorization (MF). We include in this subsection the experiments involving different variants of Matrix Factorization as well as their implementation details.

**Implementation details.** We have implemented two variants of MF: Low-Rank MF and Additive MF. The MF algorithm used in Section 5 is Low-Rank MF with rank $r = 1$. We describe the implementation details below.

- **Low-Rank MF.** Recall that in the context of our experiments, we can view each contestant as a row and each contest as a column. The score of a contestant in a contest is the entry in the corresponding row and column. A classical model of MF Koren *et al.* [2009] is factorizing $\mathbf{A} \in \mathbb{R}^{n \times m}$ as the product of two low-rank matrices $\mathbf{U}\mathbf{V}^\top$, where $\mathbf{U} \in \mathbb{R}^{n \times r}, \mathbf{V} \in \mathbb{R}^{m \times r}$ for some small $r$. Note that in our experiments, the algorithm is required to predict a new column of $\mathbf{A}$ with no known entries. Therefore, we cannot directly apply this method since the corresponding row of $\mathbf{V}$ will remain unchanged after initialization. To solve this problem, we instead predict every column with known entries in $\mathbf{A}$ and then take the average of the predictions as the prediction for the new column. We use the standard loss function that sums up the squared errors of all observed entries. We implement this method with SciPy [Jones *et al.*, 2014] and use gradient descent for a fixed number of epochs on a deterministic initialization to keep the results deterministic. We test $r = 1, 2, 5$ in this subsection.

- **Additive MF.** We also consider an additive variant of MF. For $\mathbf{x} \in \mathbb{R}^n, \mathbf{y} \in \mathbb{R}^m$, this method predicts $A_{i,j} = x_i + y_j$. Here, $x_i$ can be viewed as contestant $i$'s skill level, and $y_j$ can be interpreted as the (inversed) difficulty of contest $j$. We then use the vector $\mathbf{x}$ to make predictions. Note that this version of MF resembles QRJA in that for each of these two methods, the loss function is 0 if $A_{i,j} = x_i + y_j$ holds for the known entries. We also use the standard sum of the squared loss function and use gradient descent for a fixed number of epochs on a deterministic initialization to keep it deterministic.

**Performance experiments.** We first evaluate these variants of MF using the same metrics as in Section 5 on all datasets. The results are shown in Fig. 9. We can see that R1 MF and Additive MF

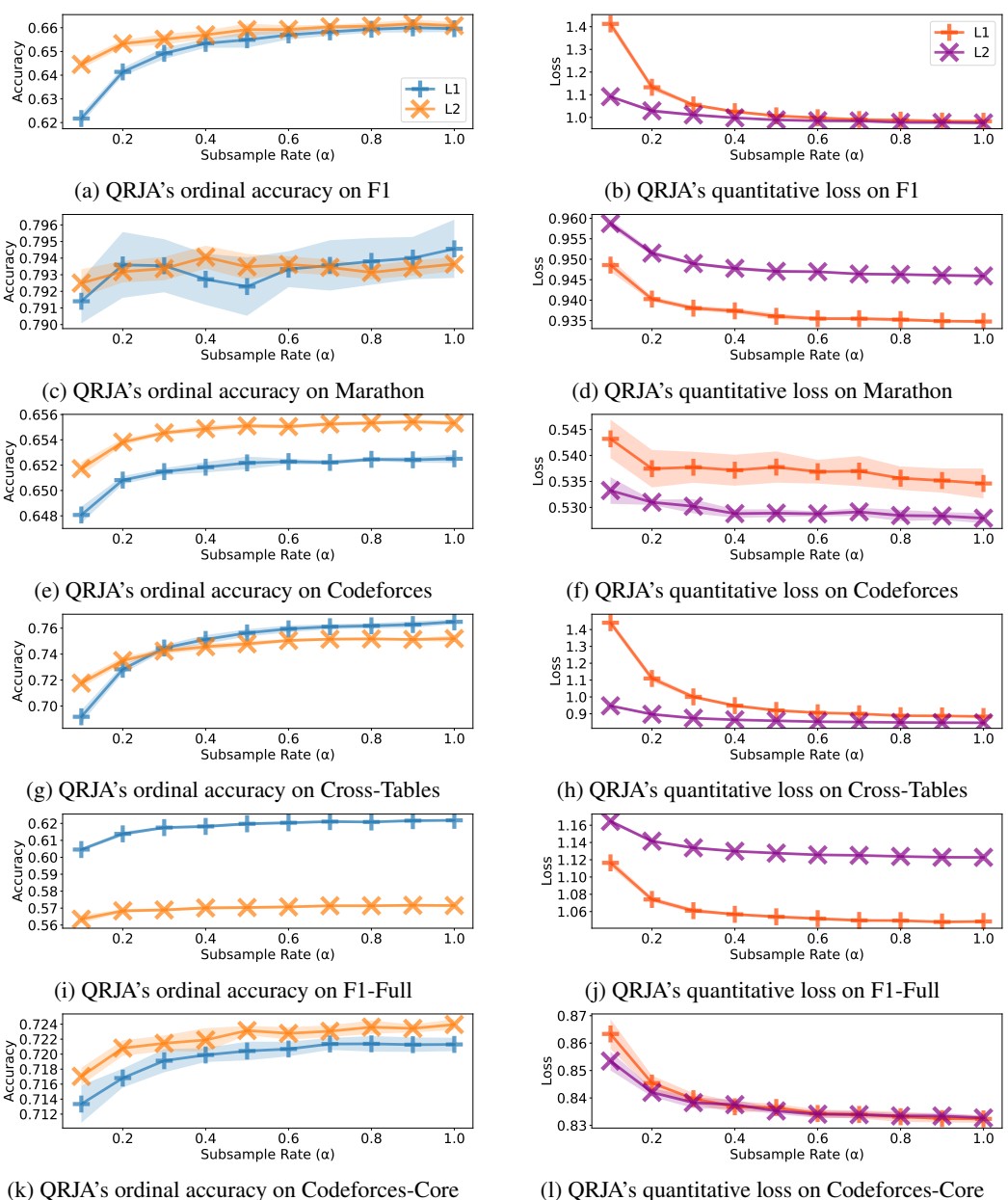

(a) QRJA's ordinal accuracy on F1

(b) QRJA's quantitative loss on F1

(c) QRJA's ordinal accuracy on Marathon

(d) QRJA's quantitative loss on Marathon

(e) QRJA's ordinal accuracy on Codeforces

(f) QRJA's quantitative loss on Codeforces

(g) QRJA's ordinal accuracy on Cross-Tables

(h) QRJA's quantitative loss on Cross-Tables

(i) QRJA's ordinal accuracy on F1-Full

(j) QRJA's quantitative loss on F1-Full

(k) QRJA's ordinal accuracy on Codeforces-Core

(l) QRJA's quantitative loss on Codeforces-Core

Figure 8: The performance of $\ell_1$ and $\ell_2$ QRJA after subsampling judgments using Algorithm 1 with equal weights for all judgments. The subsample rate $\alpha$ means $M = \lfloor \alpha m \rfloor$ in Algorithm 1. Error bars indicate the standard deviation. The results show that Algorithm 1 can reduce the number of judgments to a factor less than $1.0$ with a minor loss in performance in the used datasets. Note that errors in some figures appear large because of the small scale of the y-axis. The actual errors are small.

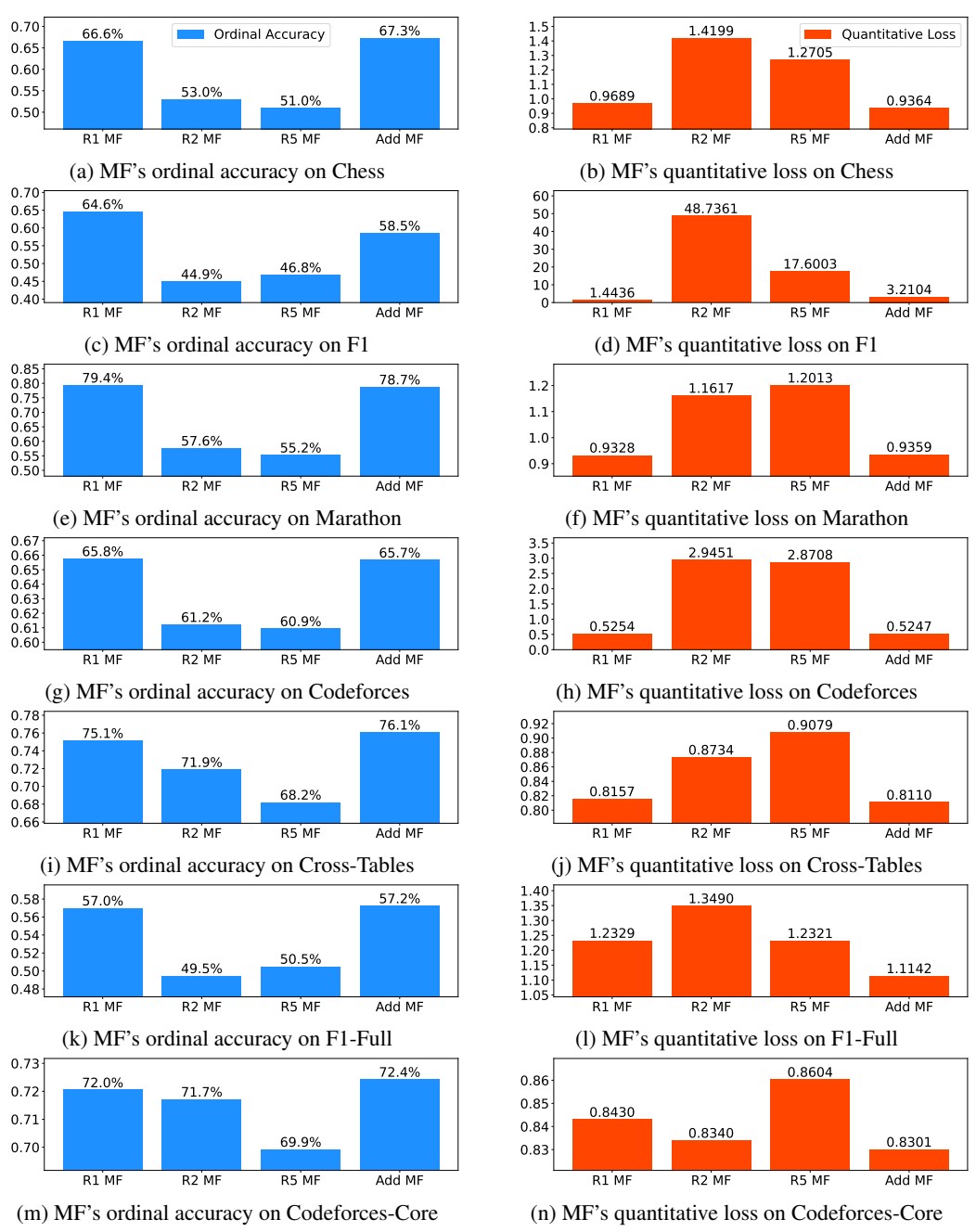

Figure 9: The performance of different variants of Matrix Factorization. The results show that R1 MF and Additive MF generally have similar performance. In contrast, R2 and R5 MF perform worse than the former.

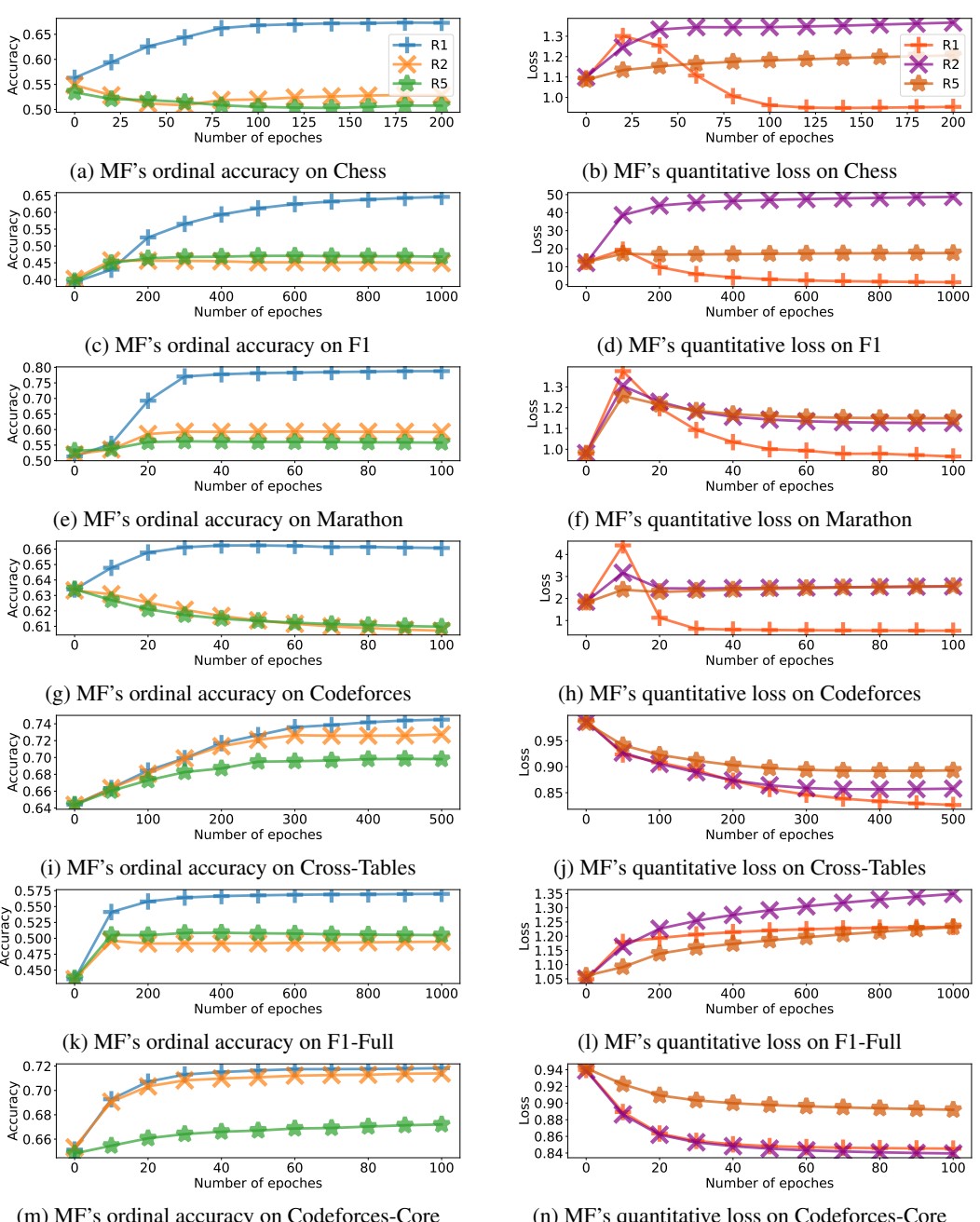

(a) MF's ordinal accuracy on Chess

(b) MF's quantitative loss on Chess

(c) MF's ordinal accuracy on F1

(d) MF's quantitative loss on F1

(e) MF's ordinal accuracy on Marathon

(f) MF's quantitative loss on Marathon

(g) MF's ordinal accuracy on Codeforces

(h) MF's quantitative loss on Codeforces

(i) MF's ordinal accuracy on Cross-Tables

(j) MF's quantitative loss on Cross-Tables

(k) MF's ordinal accuracy on F1-Full

(l) MF's quantitative loss on F1-Full

(m) MF's ordinal accuracy on Codeforces-Core

(n) MF's quantitative loss on Codeforces-Core

Figure 10: The performance of Matrix Factorization with different numbers of training epochs on all datasets. The results generally show that R1 MF outperforms R2 and R5 MF. Moreover, on some datasets, R2 and R5 MF's performance worsens as the number of training epochs increases. In contrast, R1 MF's performance improves as the number of training epochs increases.

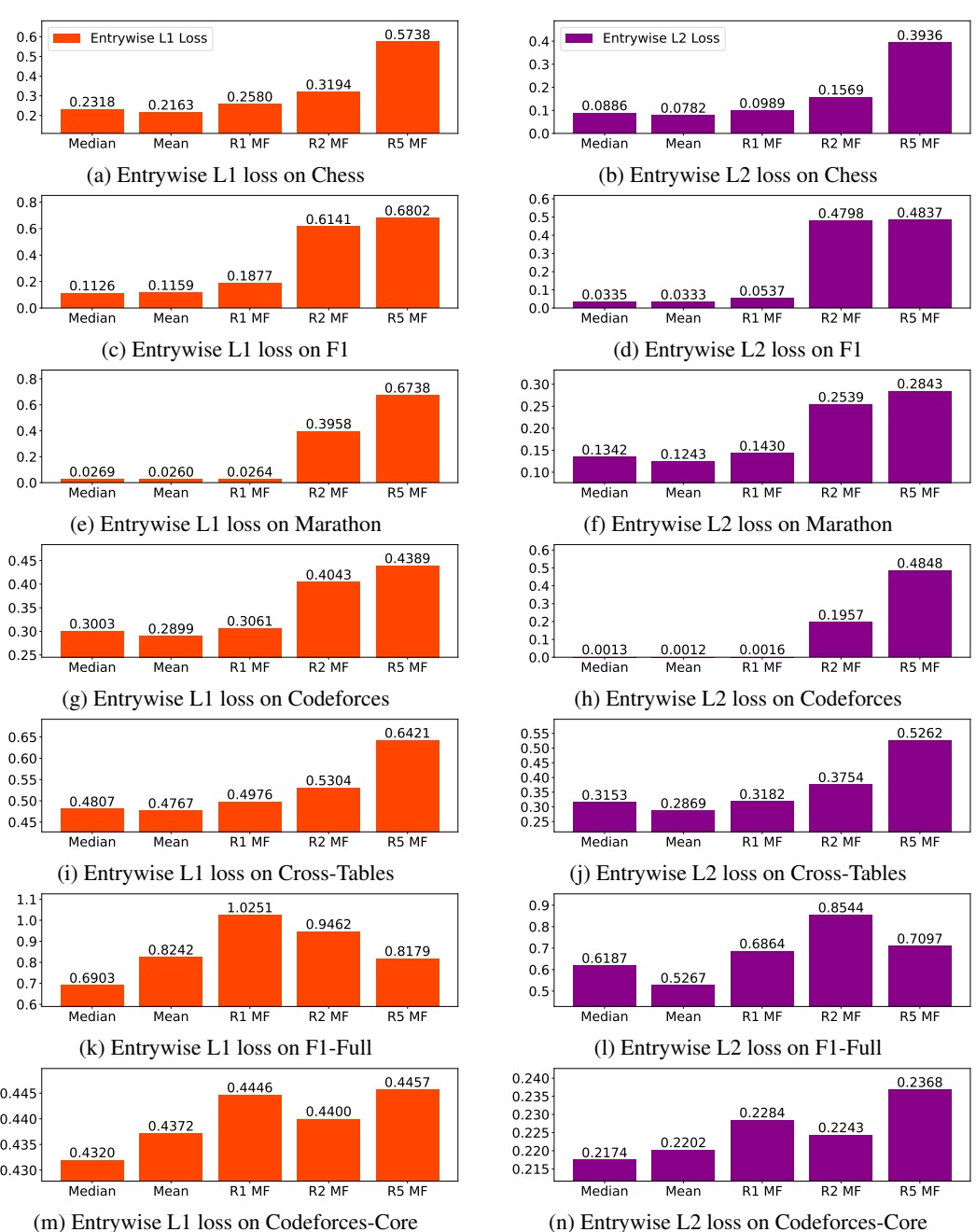

Figure 11: Entrywise L1 and L2 loss of Matrix Factorization, Mean, and Median. The results show that on most datasets, R1 MF outperforms R2 and R5 MF. The exceptions are F1-Full and Codeforces-Core. Moreover, Matrix Factorization does not have a clear advantage over Mean and Median on any dataset in terms of entrywise metrics.

generally have similar performance. In contrast, R2 and R5 MF perform worse than the former. We therefore choose to present R1 MF in Section 5.

**Low-Rank MF's performance over training.** The observation that R2 and R5 MF perform worse than R1 MF is surprising to us. To confirm this observation, we plot the performance of these variants of MF with different numbers of training epochs on all datasets. The results are shown in Fig. 10. We can see that R1 MF generally outperforms R2 and R5 MF in terms of both ordinal accuracy and quantitative loss when trained for long enough. Moreover, R1 MF's performance on both metrics generally improves as the number of training epochs increases (the only exception is quantitative loss on F1-Full). In contrast, R2 and R5 MF's performance in terms of both metrics worsens as the number of training epochs increases on Chess, F1, and Codeforces. These observed phenomena suggest that R2 and R5 MF tend to overfit the data. The problem for R1 MF is less severe.

**Experiment results on entrywise metrics.** As the metrics in Section 5 are defined in a pairwise fashion and might not be well-suited for MF, we also evaluate the performance of MF in terms of entrywise L1 and L2 loss (i.e., the average absolute and squared error of the predictions on each contestant's actual score in each contest). We also normalize each of these losses by the corresponding loss of the trivial all-zero prediction. The results are shown in Fig. 11. Note that QRJA and Additive MF are not included, because their predictions can be shifted by an arbitrary constant, and thus entrywise losses do not apply to them. We can see that in terms of entrywise L1 and L2 loss, R1 MF outperforms R2 and R5 MF on most datasets. The exceptions are F1-Full and Codeforces-Core. These two datasets are different from the other ones in that F1-Full's scores are calculated with two numbers (the number of laps finished and the finishing time) and Codeforces-Core is a dense dataset constructed from Codeforces. Therefore, on these datasets, MF with higher ranks might be more suitable than R1 MF, while on the other datasets, they tend to overfit the training data. Moreover, we note that on entrywise metrics, MF generally performs worse than Mean and Median.

**Summary of experiment results.** In summary, experiments in this subsection show that on our datasets, R1 MF and Additive MF, which are similar in performance, generally perform better than R2 and R5 MF. Therefore, we choose to include only the results of R1 MF in Section 5.

# D   Axiomatic Characterization of $\ell_p$ QRJA

We characterize $\ell_p$ QRJA by giving a set of axioms for the family of transformation functions $f$ of pairwise loss that we consider. We show that those transformation functions considered in $\ell_p$ QRJA are essentially the minimum set of functions satisfying these axioms.

Recall that for each judgment about $a$ and $b$ where $a$ is better $b$ by $y$ units, the absolute error of the prediction vector $\mathbf{x}$ on this pair is $|x_a - x_b - y|$. Using this as the loss function, we obtain the $\ell_1$ QRJA rule, which has been characterized using axioms in the context of social choice theory Conitzer *et al.* [2016]. Below we extend this characterization to $\ell_p$ QRJA for any positive rational number $p \in \mathbb{Q}_+$. Note that restricting $p$ to be rational is without loss of generality, since the output of $\ell_p$ QRJA is continuous in $p$.

We consider transforming the absolute error by a transformation function $f$ to obtain the actual pairwise loss, which is $f(|x_a - x_b - y|)$. For $\ell_p$ QRJA, the transformation function is $f(t) = t^p$. To characterize QRJA as a family of rules (for different $p \in \mathbb{Q}_+$), we give axioms for the corresponding family of transformation functions, i.e., $t^p$ for $p \in \mathbb{Q}_+$. Let $\mathcal{F}$ be a family of transformation functions.

Below are the axioms we consider:

- *Identity.* There is an identity transformation $f_0 \in \mathcal{F}$, such that $f_0(t) = t$ for any $t \geq 0$.
- *Invertibility.* For each $f_1 \in \mathcal{F}$, there is an $f_2 \in \mathcal{F}$ such that $f_1$ composed with $f_2$ is identity, i.e., for any $t \geq 0$,
$$f_1(f_2(t)) = t.$$
- *Closedness under multiplication.* For any $f_1, f_2 \in \mathcal{F}$, there exists $f_3 \in \mathcal{F}$ such that for any $t \geq 0$,
$$f_1(t) \cdot f_2(t) = f_3(t).$$

We show below that the family of transformation functions corresponding to the $\ell_p$ QRJA rules is the minimum family of functions $\mathcal{F}^*$ satisfying the above axioms. By the first axiom, the identity

transformation $f_0$ where $f_0(t) = t$ is in $\mathcal{F}^*$. (This corresponds to $\ell_1$ QRJA.) Then by the third axiom, for any $k \in \mathbb{Z}_+$, $f_0^k$ is also in $\mathcal{F}^*$, where $f_0^k(t) = t^k$. And by the second axiom, for any $k \in \mathbb{Z}_+$, $f_0^{1/k}$ is also in $\mathcal{F}^*$, where $f_0^{1/k}(t) = t^{1/k}$. This is because $f_0^{1/k}(f_0^k(t)) = t$. Finally, for any $r \in \mathbb{Q}_+$ where $r = p/q$ for $p, q \in \mathbb{Z}_+$, by the third axiom, $f_0^r = (f_0^{1/q})^p$ is in $\mathcal{F}^*$, where $f_0^r(t) = t^r$.

Note that the above argument establishes that $\mathcal{F}^*$ contains all transformation functions corresponding to QRJA, i.e.,

$$\{t^r \mid r \in \mathbb{Q}_+\} \subseteq \mathcal{F}^*.$$

Below we show the other direction, i.e., $\{t^r \mid r \in \mathbb{Q}_+\}$ satisfy the 3 axioms, and as a result,

$$\mathcal{F}^* \subseteq \{t^r \mid r \in \mathbb{Q}_+\}.$$

For $f_1(t) = t^{r_1}$, $f_2(t) = t^{r_2}$ where $r_1, r_2 \in \mathbb{Q}_+$, we have

$$f_1(t) \cdot f_2(t) = t^{r_1 + r_2},$$

where $r_1 + r_2 \in \mathbb{Q}_+$, and

$$f_1(f_2(t)) = (t^{r_2})^{r_1} = t^{r_1 \cdot r_2},$$

where $r_1 \cdot r_2 \in \mathbb{Q}_+$. This implies $\mathcal{F}^* \subseteq \{t^r \mid r \in \mathbb{Q}_+\}$. Thus $\mathcal{F}^* = \{t^r \mid r \in \mathbb{Q}_+\}$ as desired.

# E    Copyright Information for Datasets Used

The datasets used in this paper are collected from publicly available websites either manually or through an API. We provide the following information about these datasets.

- **Chess.** Copyright: © 2023 - Tata Steel Chess Tournament. Data collected is subject to the website's Terms of Conditions, available at `https://tatasteelchess.com/terms-and-conditions/`.
- **F1.** Copyright: © 2003-2024 Formula One World Championship Limited. Data collected is subject to the website's Terms of Use, available at `https://account.formula1.com/#/en/terms-of-use`.
- **Marathon.** Copyright: © 2000-2024, All Rights Reserved by MarathonGuide.com LLC. Data collected is subject to the website's Policy, available at `https://www.marathonguide.com/Policy.cfm`.
- **Codeforces.** Copyright: © 2010-2024 Mike Mirzayanov. Data collected is subject to the website's Terms and Conditions, available at `https://codeforces.com/terms`.
- **Cross-Tables.** Copyright: © 2005-2024 Seth Lipkin and Keith Smith. Data collected is subject to the website's Policy, available at `https://www.cross-tables.com/privacy.html`.

