# OpenReview forum: "Aggregating Quantitative Relative Judgments: From Social Choice to Ranking Prediction"
_NeurIPS.cc/2024/Conference — NeurIPS 2024 poster_

### Official Review · Reviewer_Z9Qp · 2024-07-02

**Soundness:** 3
**Presentation:** 3
**Contribution:** 3
**Rating:** 7
**Confidence:** 3

**Summary:**

NOTE: I have reviewed a previous version of this paper submitted to AAAI 2024. The review here is an updated version of that review reflecting the changes in the paper.

==============================

This paper explores the problem of comparing numerical judgements against each other in order to arrive at a scoring of candidates. e.g. Comparing the scores of competitors against one another, possibly when some scores are missing. The model of Quantitative Relative Judgement Aggregation accepts a set of tuples (a, b, y) which indicate that a is better than b by y units. The goal of the framework is a score for each candidate such that the difference between scores of each candidate is similar to the reported difference between candidates.

The paper narrows their framework to l_p QRJA which uses a more restricted loss function in order to ease the use of known complexity results. They show various complexity results as p changes, notably that l_p QRJA can be solved in near-linear time with bounded error for p >= 1 and that the problem is NP-hard when p <1. Experiments on several datasets show that using p = 1 and p = 2 does lead to a higher accuracy and lower loss than a number of other comparable metrics.

**Strengths:**

Novelty

The paper appears to extend an existing framework. The results expressed in the paper and the evaluation of their framework are novel while the methodology used is standard.


Quality

The paper presents several theoretical results which appear quite plausible. I am not well equipped to thoroughly check the proof of Theorem 1 but it appears to me to be quite reasonable.

The experimental results compare the developed framework across multiple datasets with several other methodologies that can be used to solve the same problem. Some of the other methodologies are more well-suited to the task than others but overall this comparison is substantial and provides a useful overview of the performance of l_p QRJA.


Clarity

The paper is well written in general. The introduction and motivating examples provide excellent intuition as to the problem domain. The framework and theory are fairly clearly described; I do feel as though the significance of the results could be highlighted more strongly but overall the paper is quite understandable.


Significance

The main result of the paper is theory showing the complexity of their l_p QRJA framework under varying conditions, backed by simulations verifying the performance of the framework. The paper is a meaningful improvement to existing work which appears roughly as impactful as most papers in top-tier conferences. There is certainly room for future work to develop based on this or for this work to be adapted into an algorithm for deployed use.

**Weaknesses:**

See previously written and lightly edited section above.

**Questions:**

Have you re-run the experiments since the previous submission that I reviewed? I ask because the numbers in Figure 4 are quite similar but do not match exactly the previous version. I cannot figure out why 2 weeks of server time would have been spent re-running the same (?) experiments previously reported.

**Limitations:**

The authors have done a good job of filling out the checklist and have addressed the limitations of the paper.

---

> ### Author Rebuttal · Authors · 2024-08-02
>
> Thank you for your recognition and thoughtful comments.
>
> We are grateful that you updated your review reflecting the changes in our paper. Compared with the AAAI 2024 version, we significantly improved our work. Notably, we strengthened our theoretical results \- our algorithm was improved from polynomial time to almost linear time. In addition, we reorganized the paper structure, re-ran some experiments, and addressed other reviewer comments.
>
> **On your question**
>
> We have re-run the experiments. This is because one of the AAAI 2024 reviewers suggested that we should “consider the average of the square of the error of the predictions”, i.e., an $\\ell\_2$ version of the quantitative loss we used. Therefore, in the subsequent revision of our work, we re-ran the experiments to incorporate this metric. See Appendix C.1 for the details.

---

> > ### Comment · Reviewer_Z9Qp · 2024-08-12
> >
> > Thank you for the response!

---

### Official Review · Reviewer_zJAB · 2024-07-07

**Soundness:** 4
**Presentation:** 3
**Contribution:** 3
**Rating:** 8
**Confidence:** 4

**Summary:**

This work defines and studies the "Quantitative Relative Judgment Aggregation" problem, which involves asking a set of judge agents to predict the performance of a set of "competitor agents" in some kind of content (e.g., a race). This task is related to recsys style collaborative filtering, ranking systems in games, and other models of preference-over-items, but the work argues this particular task is distinct and should be theoretically analyzed as a distinct setting. The paper provides several theorems about the computational complexity of QRJA and experiments using real competition data with a backtesting-style evaluation.

**Strengths:**

This paper has a number of strengths. Below, they are briefly listed. I'll also note that while the overall length of the "Weaknesses" section in this review is longer than "Strengths", overall the potential contribution here is very strong.

- Theoretical analysis of computational complexity for a new problem (variant of previous problem) + experiments to back this up.
- The paper includes several novel aspects (updating task and framing within an existing literature on social choice, the actual results)
- Code is shared, documented, and useful in providing additional clarity about experiments.
- Relatively clear paper overall. The actual presentation of model, code (supplementary materials) + experiments are all extremely clear. The explanation and presentation of Theorems 1 and 2 were reasonably clear (familiar with some of the cited social choice work, but had to refer to the provided previous work to understand the motivation for different loss functions f -- this could be a bit clearer).

In terms of significance, this work has the potential to be broadly relevant to many areas where social choice theory is relevant (i.e., many domains, as the current draft points out) and also will be of interest to readers who are familiar with this specific literature, the computational complexity results of similar methods, the implementation, etc.

**Weaknesses:**

One minor point regarding the interpretation of experimental results: the specific claim that QRJA is inherently more interpretable than MF was made a bit briefly and might benefit from additional remarks. My initial interpretation of the experimental results was: it seems like the new method introduced in the paper is about as good as MF, and has potential explainability benefits because we can walk through / "print out" the aggregation steps that took place and explain how a QRJA-using system came to the conclusion that Alice is faster than Bob. However, it seems possible that in the realm of MF one can apply various results from explainability in recommender systems (including post-hoc methods, e.g. . Concretely, I think just a brief expansion of this point would be helpful, and especially the extent to which there's a core modelling limitation with latent factors approach vs. a practical problem of 'need large embedding space to make MF good for these datasets'.

From the perspective of someone working on social choice-related theory or experimental work, I think this paper will be a strong contribution. However, one potential high-level weakness with this paper for a general audience is that readers may have some trouble understanding the general motivation for this problem, because of the mixed examples (trying to use "judgments" which are actually sensor readings from e.g. a relatively objective "race" to estimate a physical parameter of something like a car vs. trying to aggregate human judgements). While I think after spending some time with the paper, I get what the current draft is going for (something along the lines of, "social choice theory is useful in the context of these subjective aggregation problems in which some people will be weighted more or less, and is ALSO useful in estimating real "physical" models), I think this may not be totally clear.

Section 2 is helpful in this regard, but I think it could be useful to just generally clarify if the first-order goal here is to use this method to model "true rankings".

**Questions:**

Summarizing some of the points above into questions:
- Can you further clarify the explainability benefits of your approach vs MF, perhaps drawing on the provide "Examples" section
- Is it possible to provide additional clarity about the primary motivation for the work (estimating "true" / "physical" values vs. contributing to work on aggregating subjective values)
- Minor: tying the choice of loss functions back to the provided examples and motivation could make the results more exciting for a broad audience / general ML audience.

**Limitations:**

The current draft is reasonable in its claims: the theoretical analyses and experiments provided are commensurate to the claimed contributions.

---

> ### Author Rebuttal · Authors · 2024-08-02
>
> Thank you for your recognition and thoughtful comments.
>
> We are glad that you find our contribution novel and strong, and our presentation extremely clear. We are particularly delighted to see you noted and recognized our documented code for the experiments. Below, we respond to the questions raised in your review.
>
> **On the first question**
>
> QRJA and Matrix Factorization (MF) are both optimization-based methods for contest ranking prediction. QRJA can have explainability benefits over MF because its variables have clear intuitive meanings: they can be interpreted as the strength of each contestant. In contrast, the variables in MF are latent features of contests and contestants, which can be harder to interpret. We will expand the discussion of this in our paper.
>
> **On the second question**
>
> The general formulation of QRJA is motivated by the common need of aggregating quantitative relative “judgments”. These judgments are often subjective opinions from human judges, e.g.,“using 1 unit of gasoline is as bad as creating 3 units of landfill trash”. Our modeling and theoretical study of QRJA is motivated by the need of aggregating such judgments.
>
> Moreover, we observe that these relative “judgments” can also be produced by an objective process, like a contest, rather than human judges. This gives us the opportunity to explore the interplay between social choice theory and the learning-to-rank community. Therefore, we apply the QRJA rules to the problem of contest ranking prediction for further study.
>
> We will reorganize the introduction to provide additional clarity about the motivation.
>
> **On the third question**
>
> Thank you for the constructive comments.

---

> > ### Comment · Reviewer_zJAB · 2024-08-07
> >
> > Thanks for these responses. This helped to clarify my minor questions about the paper, and I stand by my original positive recommendation.

---

### Official Review · Reviewer_Fgap · 2024-07-12

**Soundness:** 4
**Presentation:** 2
**Contribution:** 2
**Rating:** 5
**Confidence:** 4

**Summary:**

The paper studies Quantitative Relative Judgment Aggregation, in which they want to learn a quantitative score for a group of alternatives that align with a series of pairwise quantitative differences between alternatives as much as possible. They extend the result in [Conitzer et al, 2016] from linear loss function to high-order loss function and show the NP-hardness to find an optimal score when the order $p < 1$. They also test their algorithms on real-world racing data and shows its supremacy towards previous methods.

**Strengths:**

+ QJRA problem is indeed interesting and important.
+ The theoretical results are sound and non-trivial.
+ Clear example demonstrate why simple methods do not work.

**Weaknesses:**

-The relationship between previous work. I don't quite see how this paper conceptually distinguishes itself from [Conitzer et al, 2016].
The two papers study similar problems under similar settings. In this case, I would rather grant the conceptual contribution of "social-choice-motivated solution concepts to the problem of ranking prediction" to their paper. Your paper indeed has its own conceptual contribution compared to the previous work, but the current claim is not precise enough.
- The motivation of the race. I feel that the whole introduction emphasizes that the paper focuses on race scenarios, yet multiple questions are unanswered: For example, why race is a reasonable representation of judgments? What are the differences between traditional scenarios that are worth a separate study? What are the real-world applications?
- The proofs and the sketches are hard to follow with too many skips and magic numbers.

**Questions:**

1. Please distinguish your paper from previous work, especially how your work is different from [Conitzer et al, 2016] and [Conitzer et al, 2015] conceptually and technically.
2. Please answer my question related to the motivation of the race in the weakness section.
3. Please specify what simple algorithm you apply for $l_2$ norm in the experiment.

**Limitations:**

Yes.

---

> ### Author Rebuttal · Authors · 2024-08-02
>
> Thank you for your thoughtful comments.
>
> We are glad that you find our problem interesting and important, and our theoretical results sound and non-trivial. Below, we respond to the questions raised in your review.
>
> **On the first question**
>
> Conitzer et al., 2015 \[1\] is a short visionary paper that proposes the abstract problem of setting numerical standards of societal tradeoffs such as “using 1 unit of gasoline is as bad as creating 3 units of landfill trash”. Conitzer et al., 2016 \[2\] axiomatically characterize a specific aggregation rule for the societal tradeoffs problem, which is mathematically equivalent to QRJA with loss function $f(x) \= x$. Zhang et al., 2019 \[3\] study the computation complexity of the specific aggregation rule characterized in \[2\].
>
> Conceptually, \[1\], \[2\] and \[3\] are all confined to computational social choice, since the societal tradeoffs problem was originally motivated by setting numerical tradeoff standards. In our work, the “relative judgments” to be aggregated can be those subjective judgments as in \[1\], \[2\] and \[3\], but they can also be produced by an objective process, like a race, rather than human judges. This gives us the opportunity to explore the interplay between social choice theory and the learning-to-rank community. In this sense, we "apply social-choice-motivated solution concepts to the problem of ranking prediction".
>
> On the technical level, only \[3\] studies computational problems related to our work. However, the techniques used by \[3\] and our work are fundamentally different. \[3\] takes a linear programming-based approach, while we need to use convex optimization techniques. We present a non-trivial reduction to the convex minimum cost flow problem and utilize recent advancements for this problem \[4\] to solve $\\ell\_p$ QRJA when $p \\geq 1$. To complement our results, we show that when $p \< 1$, $\\ell\_p$ QRJA is NP-hard by reducing from Max-Cut. None of these techniques are present in \[3\].
>
> **On the second question**
>
> We observe that the “relative judgments” can also be produced by an objective process, like races. In this sense, races are instances of judgments, rather than the representation of judgments.
>
> This specific instance of judgments is worth a separate (empirical) study because of its conceptual value of bridging the social choice and the learning-to-rank communities. That being said, our QRJA model and theoretical results are not confined to races. These contributions are also valuable within the social choice community.
>
> One direct real-world application is assigning contest ratings to a set of contestants, as illustrated in Section 2 of our paper. Besides that, our work can also be applied to the use cases of prior works on societal tradeoffs \[1-3\] like setting numerical tradeoff standards.
>
> **On the third question**
>
> $\\ell\_2$ QRJA is reducible to $\\ell\_2$-regression, which is often referred to as the linear least-square regression problem. In our experiments, we use the Python package `scipy.sparse.linalg.lsqr` to solve it (see Line 120 of the uploaded `code/qrja.py` in supplementary materials). We will clarify this.
>
> **References**
>
> \[1\] Conitzer, V.; Brill, M.; and Freeman, R. 2015\. Crowdsourcing societal tradeoffs. In Proc. of the 14th International Conference on Autonomous Agents and Multi-Agent Systems (AAMAS).
>
> \[2\] Conitzer, V.; Freeman, R.; Brill, M.; and Li, Y. 2016\. Rules for choosing societal tradeoffs. In Proc. of the 30th AAAI Conference on Artificial Intelligence (AAAI).
>
> \[3\] Zhang, H.; Cheng, Y.; and Conitzer, V. 2019\. A better algorithm for societal tradeoffs. In Proc. of the 33rd AAAI Conference on Artificial Intelligence (AAAI).
>
> \[4\] Chen, L.; Kyng, R.; Liu, Y. P.; Peng, R.; Gutenberg, M. P.; and Sachdeva, S. 2022\. Maximum flow and minimum-cost flow in almost-linear time. In Proc. of the IEEE 63rd Symposium on Foundations of Computer Science (FOCS).

---

> > ### Comment · Reviewer_Fgap · 2024-08-08
> >
> > Thank you for your response. I have one additional question related to your response on question 1.
> >
> > In the second paragraph, you state that "(relative judgments to be aggregated) can also be produced by an objective process, like a race, rather than human judges" and then "This gives us the opportunity to explore the interplay between social choice theory and the learning-to-rank community". Could you deliberate it bit more on how the first statement logically leads to the second? Is it because that the subjective human judges does not fit into learning-to-ranking problems? Or are there other reasons?

---

> > > ### Author Response · Authors · 2024-08-08
> > >
> > > Thank you for your reply.
> > >
> > > Generally speaking, in social choice, people typically consider subjective opinions and judges as inputs, e.g., voting. Conversely, in the learning-to-rank community, the inputs are usually more objective, e.g., ranking web pages in response to a search query. The difference in subjectiveness has led to the difference in the focus of these two communities. For example, the learning-to-rank community is less concerned about strategic aspects of voting. By considering a social-choice-motivated problem with objective inputs typically seen in the learning-to-rank community, we explore the interplay between them.

---

> > > > ### Author Response · Authors · 2024-08-09
> > > >
> > > > Another difference between the communities is the degree to which aggregate rankings need to be justified, be transparent, follow a clear and simple pre-specified rule, satisfy certain axioms, etc. These aspects tend to be especially important in the social choice community; in the learning-to-rank community, it varies, but at least in some cases they do not matter too much.  For example, in many content recommendation settings, what matters primarily is that the results are "good" by some metric, such as that they contain what the user is searching for early in the results. We think the bridging role played by our current paper can also be helpful in intermediate settings, such as aggregating the results of races or contests, where we don't need to worry about the races somehow being strategic agents, but for fair evaluation we still want the results to be based on some clear interpretable rule.

---

> > > > > ### Comment · Reviewer_Fgap · 2024-08-12
> > > > >
> > > > > I appreciate your detailed discussion. I am more familiar with social choice than learning to rank, and your explanation answers my questions well. I will be more positive about this paper if you can add this discussion to the edited version of this paper. Also, I feel that while ranking objective judgments makes sense, racing data itself is not persuasive enough. I think you can discuss more applications in the motivation and make racing a running example (as you already do in the paper).

---

> > > > > > ### Author Response · Authors · 2024-08-12
> > > > > >
> > > > > > Thank you for your reply. We will incorporate this discussion in the revised version of the paper. We will also add more examples to show the broader applicability of ranking objective judgments, while continuing to use the racing scenario as a running example.

---

### Official Review · Reviewer_iMha · 2024-07-13

**Soundness:** 3
**Presentation:** 2
**Contribution:** 3
**Rating:** 6
**Confidence:** 3

**Summary:**

The paper generalizes relative quantitative judgments by Conitzer et al., which aims to aggregate judgments on the relative quality of different candidates from various sources and applies it to a learning-to-rank setting.  The authors introduce new aggregation rules QRJA that tries to assign vector $x_1,\dots x_n$ that minimizes error
$$\sum_i w_i f(|x_{a_i}-x_{b_i}-y_i|)$$
from collection of quantitative relative judgement $J_i = (a_i,b_i,y_i)$ where $f$ is some increasing function.

The paper mostly focuses on $f(t)$ being a monomial $t^p$.  They provide a near-linear time algorithm to solve the above optimization problem if $p\ge 1$ and NP-hardness results when $p<1$.  Empirically, the paper validates the effectiveness of QRJA-based methods through some simple baselines, (mean, median, Borda rule, matrix factorization, and KY rule) on on real-world datasets, including chess, Formula 1, marathons, and programming contests.

**Strengths:**

+) The combination of social choice and ranking prediction is pretty neat.
+) The dichotomy results on the degree of the monomial is intuitive and interesting.

**Weaknesses:**

1. The technical writing is not very friendly or informative.
- It seems the optimization problem can be reduced to a sparse $p$-norm regression problem.  Is there an off-shelf method to solve it?  Why is it necessary to consider the dual program?
- The proof of Lemma 1 is not clear to me.  Which results in Chen et al. 2022 are you used to solve Equation (4) (e.g., Theorem 10.14)?  Moreover, the author should provide an explicit statement of the reduction.

2.  I feel the empirical results should be compared to more relevant literature.
  - Given the long history of rank aggregation or ranking system, I feel the evaluation in the paper is insufficient.  A rating system is an algorithm that adjusts a players rating upwards after each win, or downwards after each loss.  Some notable rating systems used in practice include Harkness, Elo, Glicko, Sonas, TrueSkill, URS, and more.   As ordinal accuracy in the paper is also derived from pairwise comparisons, it seems reasonable to apply Elo and other methods.
  - Another related area may be item response theory, where contestants have expertise and events have difficulty.

3.  The presentation should be more cogent
- It is not clear to me how the model conceptually addresses issues in those motivating examples (Examples 1 to 3).
- The empirical results only test QRJA under $p = 1,2$, and provide little discussion on the choice of $p$.  Note that standard gradient descent should solve the optimization problem for all $p\ge 1$.

**Questions:**

Can you address the first point in weakness of Theorem 1?
- why dual program
- Limitation of using $p$-norm regression
- Reduction to Chen et al. 2022

**Limitations:**

The paper addresses the limitations.

---

> ### Author Rebuttal · Authors · 2024-08-01
>
> Thank you for your recognition and thoughtful comments.
>
> We are glad that you find the topic of our work neat and our theoretical dichotomy results intuitive and interesting. Below, we respond to the questions raised in your review.
>
> **On the first point in Weaknesses**
>
> $\\ell\_p$ QRJA is indeed a special case of sparse $p$-norm regression, but it has additional structure: for $\\ell\_p$ QRJA, the matrix $A$ in $\\min \\|Ax-z\\|\_p$ has exactly two nonzero entries per row and they sum to $0$. This is crucial because it allows the dual of $\\ell\_p$ QRJA to correspond to a flow problem.
>
> Without this additional structure, applying the state-of-the-art algorithm for sparse $p$-norm regression \[1\] would result in an $\\Omega(m \+ n^{\\omega})$ runtime for $\\ell\_p$ QRJA, where $\\omega \\geq 2$ is the matrix multiplication exponent. This is significantly slower than almost linear time.
>
> With this additional structure, we can solve $\\ell\_p$ QRJA in almost-linear time using Theorem 10.13 of \[2\]: In Equation (4), one can view each entry $f\_e$ in $\\mathbf{f}$ as the directed flow on an edge $e$, and the optimization constraints as flow conservation constraints. For edge $e$, its contribution to the total cost is $|f\_e|^q \- z\_e f\_e$, and the total cost is edge-separable and convex. This allows us to use Theorem 10.13 of \[2\].
>
> We will clarify this.
>
> **On the second point in Weaknesses**
>
> We agree that many other approaches deserve mention in this context, including Elo and other methods. In our work, we had to select a subset of these methods to compare with QRJA. We chose Mean and Median due to their straightforwardness, Borda and Kemeny-Young from the social choice literature, and Matrix Factorization from the machine learning literature.
>
> **On the third point in Weaknesses**
>
> QRJA addresses issues in the motivating examples by considering the relative performance of contestants rather than their absolute performance in each contest. More specifically:
> * Example 1: Even if a contestant only participates in “easy” races, QRJA can better avoid over-rating their strength by using the relative performance data of other contestants in those races.
> * Example 2: If past data shows that Alice consistently beats Bob, and Bob consistently beats Charlie, the QRJA model will predict that Alice runs faster than Charlie, as it tends to be consistent with previous relative judgments.
> * Example 3: Although the Boston race indicates that Charlie is slightly faster than Bob, the other two races suggest that Bob is faster than Charlie by a large margin. To minimize inconsistencies across all judgments, QRJA will predict that Bob is faster than Charlie.
>
> We focus on $\\ell\_1$ and $\\ell\_2$ QRJA because the almost-linear time algorithm for general values of $p \\geq 1$ relies on galactic algorithms for $\\ell\_p$ norm mincost flow \[2\]. While standard gradient descent works for general values of $p$, its running time is too slow on large datasets like Codeforces and Cross-Tables. We acknowledged this in the conclusion: “An interesting avenue for future work would be to develop fast (e.g., nearly-linear time) algorithms for $\\ell\_p$ QRJA with $p \\neq 1, 2$ that are more practical, and evaluate their empirical performance.” We will further clarify this.
>
> **References**
>
> \[1\] Bubeck, S.; Cohen, M. B.; Lee, Y. T.; and Li, Y. 2018\. An homotopy method for $\\ell\_p$ regression provably beyond self-concordance and in input-sparsity time. In Proc. of
> the 45th annual ACM Symposium on Theory of Computing (STOC).
>
> \[2\] Chen, L.; Kyng, R.; Liu, Y. P.; Peng, R.; Gutenberg, M. P.; and Sachdeva, S. 2022\. Maximum flow and minimum-cost flow in almost-linear time. In Proc. of the IEEE 63rd Symposium on Foundations of Computer Science (FOCS).

---

> > ### Comment · Reviewer_iMha · 2024-08-12
> >
> > Thanks for these responses. I maintain my original positive recommendation.

---

### Author Rebuttal · Authors · 2024-08-02

We thank all reviewers for taking the time to read our paper and provide thoughtful comments.

We are delighted to learn that the reviewers find our topic “interesting and important” (Fgap), and the “combination of social choice and ranking prediction” “neat” (iMha). In addition, we are glad that reviewers judged our theoretical results to be “sound and non-trivial” (Fgap), our empirical evaluation “substantial” (Z9Qp), and our potential contribution “very strong” (zJAB).

Below we provide detailed responses to each reviewer’s questions.

---

### Decision · Program_Chairs · 2024-09-25

**Decision:**

Accept (poster)

**Comment:**

The paper generalizes the quantitative relative judgments aggregation problem (QRJA) proposed by Conitzer et al. (2015, 2016), bridging voting theory and learning-to-rank problems. There is a latent “true” ranking of the candidates to be learned based on a set of judgments (a, b, y) indicating that candidate a is better than candidate b by y units. The paper introduces new aggregation rules for QRJA for high-order loss functions and tests their algorithms on real-world data.

The review team agreed that the problem is interesting and important and the paper presents a sound theoretical analysis and clear experiments. The reviewers pointed out several aspects that could be improved, including the presentation of the technical details, positioning of the paper within the literature (highlighting better the conceptual contributions), and discussing and comparing against related empirical works (see e.g., the suggestions and comments by Reviewer iMha, zJAB).

Overall, the review team, including myself, view positively the contributions of the paper. I believe most of the concerns can be sufficiently addressed in a revision (given the information provided by the authors in their rebuttal).